# PREDICTING PERTURBATION TARGETS WITH CAUSAL DIFFERENTIAL NETWORKS

## ABSTRACT

Rationally identifying variables responsible for changes to a biological system can enable myriad applications in disease understanding and cell engineering. From a causality perspective, we are given two datasets generated by the same causal model, one observational (control) and one interventional (perturbed). The goal is to isolate the subset of measured variables (e.g. genes) that were the *targets* of the intervention, i.e. those whose conditional independencies have changed. Knowing the causal graph would limit the search space, allowing us to efficiently pinpoint these variables. However, current algorithms that infer causal graphs in the presence of unknown intervention targets scale poorly to the hundreds or thousands of variables in biological data, as they must jointly search the combinatorial spaces of graphs *and* consistent intervention targets. In this work, we propose a causality-inspired approach for predicting perturbation targets that decouples the two search steps. First, we use an amortized causal discovery model to separately infer causal graphs from the observational and interventional datasets. Then, we learn to map these paired graphs to the sets of variables that were intervened upon, in a supervised learning framework. This approach consistently outperforms baselines for perturbation modeling on seven single-cell transcriptomics datasets, each with thousands of measured variables. We also demonstrate significant improvements over six causal discovery algorithms in predicting intervention targets across a variety of tractable, synthetic datasets.

## 1 INTRODUCTION

Cells form the basis of biological systems, and they take on a multitude of dynamical states throughout their lifetime. In addition to natural factors like cell cycle, external perturbations (e.g. drugs, gene knockdown) can alter a cell's state. While perturbations can affect numerous downstream variables, identifying the root causes, or *targets*, that drive these transitions has vast therapeutic implications, from cellular reprogramming (Cherry & Daley, 2012) to mechanism of action elucidation (Schenone et al., 2013). Large-scale experiments (Replogle et al., 2022) have attempted to map the effects of perturbing individual genes on single cells, and *in-silico* approaches (Roohani et al., 2023; Lotfollahi et al., 2023) have been designed for the "forward" inference task of predicting the post-perturbation expression of each gene. In principle, these models can be used within an active learning framework to suggest perturbation targets (Huang et al., 2023; Zhang et al., 2023a). However, the number of inference calls required scales exponentially with the size of the candidate set (e.g. drugs with off-target effects), rendering these approaches impractical for larger search spaces. These models' predictive performance as oracles has also been called into question by subsequent works (Kernfeld et al., 2023; Ahlmann-Eltze et al., 2024), highlighting the difficulty of this forward approach. Finally, there are limited training data for combinatorial perturbations: the most widely used dataset only contains around a hundred pairs (Norman et al., 2019).

The "reverse" strategy – of directly predicting targets from perturbation data – alleviates these problems to an extent. Towards this end, it is common to assume (sometimes implicitly) that the data were generated by a structured causal model, and the perturbation targets can be directly inferred from the changes to this model. While the effects of perturbations are highly contextual, specific to the cell lines in which they are applied (Nadig et al., 2024), existing methods rely on non-specific, data-mined knowledge graphs as the backbones for their causal structures (Cosgrove et al., 2008; Gonzalez et al., 2024). It is hard to quantify the extent to which these graphs are relevant to each

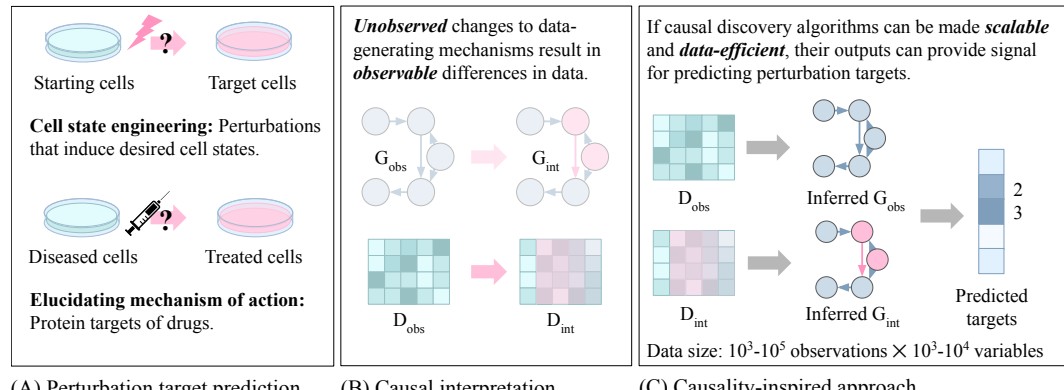

(A) Perturbation target prediction    (B) Causal interpretation    (C) Causality-inspired approach

Figure 1: (A) Biological applications. (B) Differences between observation dataset $D_{\text{obs}}$ and interventional dataset $D_{\text{int}}$ can be attributed to changes in the underlying causal model $G$. (C) We use an amortized causal discovery module to obtain features that represent $G_{\text{obs}}$ and $G_{\text{int}}$. Then we train a supervised classifier to predict the unknown intervention targets.

setting, as the quality and focus of assays have changed over time, and interactions present in one cell type may not exist in another (Huttlin et al., 2021).

Our goal is to predict the perturbation targets that translate one set of cells into another, while inferring the underlying causal structures directly from the data. We formalize this task within a causality framework. Specifically, given an *observational* dataset (e.g. gene expression of control cells) and an *interventional* dataset (e.g. perturbed cells) generated by the same system (e.g. cell type), we would like to identify the variables (e.g. genes) whose data-generating mechanisms are different between the two settings. Causal discovery algorithms have been designed to solve this exact problem, on interventional data with unknown targets (Squires et al., 2020; Brouillard et al., 2020; Hägele et al., 2023). These algorithms operate jointly over the spaces of graphs and intervention target sets. Due to this large search space, these algorithms are unable to scale to the thousands of variables and hundreds of thousands of observations in modern transcriptomics data, while simultaneously being robust to as few as tens of observations *per* interventional setting (Nadig et al., 2024).

To address unknown intervention target prediction in context of biological perturbations, we propose causal differential networks (CDN). Drawing upon recent advances in scalable and data-efficient causal discovery (Wu et al., 2024), CDN first "featurizes" the observational and interventional datasets in terms of their predicted causal graphs, using a pretrained causal discovery module. These paired, edge-level representations are then used as input to an axial attention-based classifier (Ho et al., 2020), which is supervised by ground truth intervention targets (on either real or synthetic data). We thoroughly evaluate CDN on both real transcriptomic data and in synthetic settings. CDN outperforms the state-of-the-art in perturbation modeling on the five largest Perturb-seq datasets at the time of publication (CRISPRi, Replogle et al. (2022); Nadig et al. (2024)), and two Sci-Plex datasets (chemical perturbations, McFaline-Figueroa et al. (2024)), without using any external knowledge. In fact, CDN is the only model that consistently ranks the ground truth perturbation targets higher than would be expected by random. Moreover, CDN is able to maintain decent performance even on unseen cell lines, demonstrating its potential to generalize. Finally, on synthetic settings, CDN outperforms causal discovery algorithms for unknown intervention targets. To conclude, our contributions are three-fold.

1. We propose causal differential networks (CDN), a causal discovery-based approach for perturbation target prediction on transcriptomic data, which achieves the state-of-the-art on genetic and chemical perturbation datasets.
2. We demonstrate that CDN outperforms current causal discovery approaches for unknown intervention target prediction for both hard and soft interventions in diverse synthetic data.
3. Finally, we have prepared seven high-quality transcriptomics datasets, to serve as a benchmark for future machine learning studies.

## 2 BACKGROUND AND RELATED WORK

### 2.1 IDENTIFYING PERTURBATION TARGETS

The rise of large-scale perturbation screens (Replogle et al., 2022) has enabled machine learning approaches for predicting perturbation targets. One line of work focuses on active learning to reduce experimental costs (Zhang et al., 2023a; Huang et al., 2023), but these models require inference calls to an oracle model for each candidate perturbation. Given that human cells express over ten thousand genes, and drugs may impact multiple targets, the search space is enormous. Moreover, subsequent works have found that naive baselines (mean of the training set, linear regression) outperform these approaches at their intended task (Kernfeld et al., 2023; Ahlmann-Eltze et al., 2024; Märtens et al., 2024). In contrast, perturbation targets can be predicted directly from data, using a variety of domain knowledge (Cosgrove et al., 2008; Gonzalez et al., 2024; Roohani et al., 2024). However, these external data were collected from highly inhomogenous sources, which may be inconsistent with the data at hand.

### 2.2 CAUSAL DISCOVERY WITH UNKNOWN INTERVENTION TARGETS

Perturbation target prediction can be formalized as a causal discovery problem, using data generated under interventions with unknown targets. Specifically, a causal graphical model is a directed graph $G = (V, E)$, where nodes $i \in V$ map to random variables $X_i \in X$, and edges $(i, j) \in E$ represent relationships from $X_i$ to $X_j$. There are a number of common assumptions that relate $G$ to the data distribution $P_X$, which we defer to works such as Spirtes et al. (2001); Yang et al. (2018); Zhang et al. (2023b), since the identifiability of any particular system is not the focus of this paper.

Data generated directly from $P_X$ are known as *observational* data. A causal graphical model allows us to perform *interventions* by assigning new conditionals

$$P(X_i \mid X_{\pi_i}) \leftarrow \tilde{P}(X_i \mid X_{\pi_i}), \tag{1}$$

where $\pi_i$ denotes the parents of node $i$ in $G$. *Hard* interventions remove all dependence between $X_i$ and $\pi_i$, while *soft* interventions maintain the relationship with a different conditional. We denote the joint *interventional* distribution as $\tilde{P}_X$. Given an observational dataset $D_{\text{obs}} \sim P_X$ and an interventional dataset $D_{\text{int}} \sim \tilde{P}_X$, our goal is to predict the set of nodes $I$ for which

$$P(X_i \mid X_{\pi_i}) \neq \tilde{P}(X_i \mid X_{\pi_i}), \forall i \in I. \tag{2}$$

This task has been well-studied in the causality literature, often as a sub-problem or setting while inferring graph $G$ from $D_{\text{obs}}$ and (multiple) $D_{\text{int}}$. Ke et al. (2019) first proposed a discovery algorithm for unknown interventions, on discrete data of up to 50 variables. Jaber et al. (2020) provides identifiability guarantees for soft interventions with unknown targets and proposes a constraint-based algorithm, which requires an exponential number of conditional independence tests with respect to the number of variables. Continuous causal discovery algorithms that are consistent despite unknown interventions include DCDI (Brouillard et al., 2020) and BACADI (Hägele et al., 2023), which treat intervention target identities as learnable parameters of a generative model over $P_X$. DCDI uses the Gumbel-Softmax approximation (Jang et al., 2017) to learn these categorical variables, while BACADI takes a Bayesian approach, with gradient-based methods for efficient posterior estimation (Liu & Wang, 2016). Finally, Varici et al. (2022) and Yang et al. (2024) frame intervention target prediction as a separate task, similar to this paper. However, the former strictly assumes linearity, while the latter requires data from multiple environments, which are not available here.

### 2.3 AMORTIZED CAUSAL DISCOVERY ALGORITHMS

The majority of causal discovery algorithms operate on one or more datasets $D$, which correspond to a single graph $G$ (Spirtes et al., 1995; Hauser & Bühlmann, 2012; Mooij et al., 2020; Brouillard et al., 2020). These algorithms must be fit or run from scratch for each data distribution – a potential challenge in low-resource scenarios, where there are too few observations of too many variables. Recently, amortized causal discovery algorithms have addressed causal structure learning as a *supervised* machine learning problem. Using large numbers of synthetic datasets, generated by (known) synthetic graphs, a neural network is trained to map $D$ directly to $G$ (Ke et al., 2022; Lorch et al.,

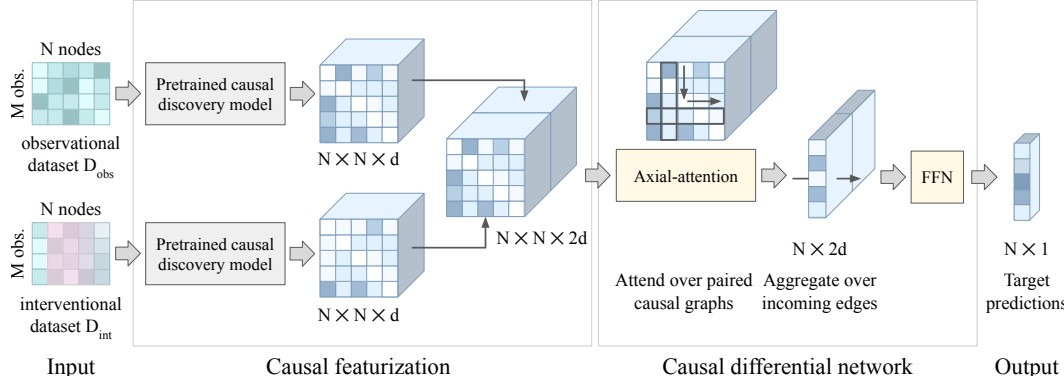

Figure 2: Model overview. The observational and interventional datasets are featurized using a pre-trained, amortized causal discovery model. The pair of graph representations are concatenated and input to an axial attention model. After the 2D attention, representations are aggregated over incoming edges to obtain binary, node-level predictions. CDN is trained with a classification objective, over synthetic and/or real data. Gray modules are frozen. Yellow modules are learned.

2022; Wu et al., 2024). Two immediate benefits are fast runtimes and robustness to aleatoric noise. In this work, we leverage the framework proposed by Wu et al. (2024), as it has been benchmarked on large, transcriptomics data (Chevalley et al., 2022).

## 3 METHODS

Our model is composed of two modules: a frozen *causal featurizer*, and a learned *differential network*, which we refer to jointly as CDN (Figure 2). The causal featurizer is a pretrained, amortized causal discovery model (Wu et al., 2024) that runs efficiently on up to a thousand nodes (Section 3.1), while the differential network is an axial-attention based classifier (Ho et al., 2020) that predicts the nodes whose data-generating mechanisms have changed (Section 3.2).

### 3.1 CAUSAL FEATURIZER

The causal featurizer takes as input datasets $D_{\text{obs}}$ and $D_{\text{int}}$ containing samples of $N$ variables, generated from observational and interventional settings. For example, $D_{\text{obs}}$ may correspond the gene expression matrix of non-targeting control cells; $D_{\text{int}}$ is the gene expression of the same type of cells, subject to a perturbation with an unknown target; and $N$ is the number of genes detected.

We use an amortized causal discovery model to obtain features $h_{\text{obs}}, h_{\text{int}} \in \mathbb{R}^{N \times N \times d}$, which reflect the $N \times N$ adjacency matrices of causal graphs that may have generated each dataset. Specifically, to featurize each dataset, we follow the three steps proposed in Wu et al. (2024).

1. Given a dataset $D$, smaller batches are constructed by sub-sampling both examples and variables. Heuristics like pairwise correlation are used to select variables which are likely to have causal relationships, to minimize unnecessary computation.
2. Two sets of input features are generated from these batches: *global* summary statistics like correlation, computed over all variables, and *marginal* estimates, the outputs of classical causal discovery algorithms (Spirtes et al., 1995) run on small subsets of variables.
3. Finally, these two sets of features are input to a neural network ("aggregator"), which was trained to map them into causal graphs on synthetic datasets.

We run this procedure on $D_{\text{obs}}$ and $D_{\text{int}}$ independently, treating both as "observational" datasets. To obtain the graph representations for each dataset, we extract the aggregator's last layer hidden representations before they are collapsed into binary edge predictions, yielding $h_{\text{obs}}$ and $h_{\text{int}}$. Finally, we concatenate $h_{\text{obs}}$ and $h_{\text{int}}$ along the hidden dimension, to obtain a paired graph representation of size $N \times N \times 2d$.

## 3.2 DIFFERENTIAL NETWORK

Given the graph representation, the differential network predicts which nodes were intervened upon (Figure 2, right). Its architecture is composed of an axial-attention layer, followed by a linear projection. Following SEA, the axial attention layer attends separately along the rows and columns of the graph's adjacency matrix. This operation is equivalent to self-attention along all nodes in the "outgoing edge" direction, with the "incoming edges" as a batch dimension, followed by the opposite. We use pre-layer normalization on each self-attention, followed by dropout and residual connections:

$$h = h + \text{Dropout}(\text{Self-Attn}(\text{LayerNorm}(h))). \tag{3}$$

After the axial attention, we mean over all "incoming edges" and linearly project to binary node-level predictions, where a normalized 1 indicates an intervention target, and 0 indicates otherwise.

## 3.3 IMPLEMENTATION AND TRAINING DETAILS

For the causal featurizer, we used a frozen, pretrained aggregator from (Wu et al., 2024), which corresponded to the FCI-based marginal estimates (Spirtes et al., 1995). The differential network was implemented with one axial attention layer, following the same architecture as the pretrained aggregator, with twice the model dimension (paired graphs). We swept over the number of layers (1-4) on synthetic data and found that a single layer performed the best. In terms of model design, we ablated replacing the attention architecture with a multi-layer perceptron (no communication between nodes, Table 4), as well as other components (Table 7) and summary statistics (Table 8). Due to the large graph sizes (adjacency matrices up to $1000^2$), standard 1D attention was intractable. Following SEA, we used inverse covariance as the global statistic on synthetic data, and correlation on transcriptomics data.

We trained separate differential networks on synthetic and real data. On synthetic data, we used a batch size of 16 and the AdamW optimizer (Loshchilov & Hutter, 2019) with a learning rate of 1e-4 (identical to SEA). On real data, we started from the synthetic CDN checkpoint, and then used a batch size of 2 with a learning rate of 1e-5, due to the larger graph sizes. All models took around 4-10 hours to train on a single A6000 GPU.

## 3.4 THEORETICAL CONTEXT

This paper primarily describes a causality-inspired approach towards perturbation target prediction. Contrary to causal discovery algorithms that are self-contained and provably sound under standard assumptions, we cannot "prove" that a pretrained model extracts correct graphs on real data, as the true graphs are unknown. Our evaluation focuses solely on perturbation target prediction, as the ground truth is known, by (experimental) design. To contextualize the predictions of our model, we provide brief sketches with regards to our model's capacity to map paired graph features to intervention targets. We defer formal claims to Appendix A.

**Claim 3.1** (Hard interventions, informal). *Given a causal graphical model $G = (V, E)$ and intervention targets $I \subseteq V$, let $E'$ denote the adjacency matrix of mutilated graph $G'$ after hard intervention on $I$. An axial attention layer is well-specified to map $(E, E')$ to $I$.*

**Claim 3.2** (Soft interventions, informal). *Let $G$ be a graphical causal model associated with data distribution $P_X$, and let $\tilde{P}_X$ be the interventional distribution after soft intervention on $I \subsetneq V$. Let $R, R'$ denote the correlation matrices and let $\Sigma, \Sigma'$ denote the covariance matrices of $X, \tilde{X}$. An axial attention layer is well-specified to map $(R, R', \Sigma, \Sigma')$ to $I$.*

## 4 EXPERIMENTS AND RESULTS

We evaluated CDN on seven transcriptomics datasets, with comparisons to state-of-the-art models for these applications (Section 4.1). Note that these baselines all use various domain knowledge to inform their predictions. For completeness, we also benchmarked CDN against multiple causal discovery algorithms for unknown interventions in variety of controlled settings (Section 4.2).

Table 1: Final biological dataset statistics. "gw" and "es" are two independent screens with different perturbations.

| Type | Cell line | # Perts | # Cells |
|---|---|---|---|
| Genetic | K562 gw | 1767 | 492,096 |
| | K562 es | 1060 | 213,552 |
| | RPE1 | 961 | 179,696 |
| | HepG2 | 627 | 79,309 |
| | Jurkat | 1014 | 174,698 |
| Chemical | A172 | 3 | 18,196 |
| | T98G | 3 | 13,126 |

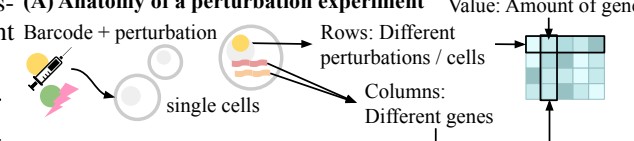

**(A) Anatomy of a perturbation experiment**

Barcode + perturbation → single cells → Rows: Different perturbations / cells

Value: Amount of gene

Columns: Different genes

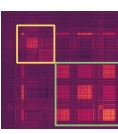

**(B) Data splitting pipeline**

1. Filter perturbations
All perturbations
10+ DE genes
Target is DE

2. Cluster by log-fold change

3. Split perturbations
*Seen* cell lines: Split by cluster
*Unseen* cell lines: same test, train from different cell lines.

Figure 3: Data illustration and splitting procedure.

## 4.1 BIOLOGICAL EXPERIMENTS

**Datasets** We evaluate CDN on five Perturb-seq (Dixit et al., 2016) datasets (genetic perturbations) from Replogle et al. (2022) and Nadig et al. (2024); as well as two Sci-Plex (Srivatsan et al., 2020) datasets (chemical perturbations) from McFaline-Figueroa et al. (2024). Each dataset is a real-valued matrix of gene expression levels: the number of examples $M$ is the number of cells, the number of variables $N$ is the number of genes, and each entry is a log-normalized count of how many copies of gene $j$ was measured from cell $i$ (Figure 3A). In Perturb-seq datasets, we aim to recover the gene whose promoter was targeted by the CRISPR guide, and in Sci-Plex datasets, we aim to identify the gene that corresponds to the drug's intended target.

To ensure high quality labels, we filtered perturbations to those that induced over 10 differentially-expressed genes (statistically significant change, compared to control), of which the true target should be present. This is to exclude perturbations with insufficient cells (low statistical power), with minimal to no effect (uninteresting), and those that did not achieve the desired effect (CRISPR efficacy is not guaranteed). For evaluation, we limited the set of candidate targets to the top 1000 differentially expressed genes, ranked by log-fold change per perturbation. If fewer genes were differentially expressed, we randomly sampled additional candidates until we reached a minimum of 100 genes. Finally, we also stratify genetic perturbations based on whether the target is trivially identifiable as the gene with the largest log-fold change ("trivial" vs. "non-trivial"). This is because genetic perturbations are highly specific by design, but also constitute the largest single-cell perturbation datasets. The final data statistics are shown in Table 1, with additional details in Appendix B.1.

**Evaluation** We consider two splits: seen and unseen cell lines (Figure 3B). In the former, models may be trained on approximately half of the perturbations from each cell line, and are evaluated on the unseen perturbations. In the latter, we hold out one cell line at a time, and models may be trained on data from the remaining cell lines. Note that not all baselines can be evaluated on unseen cell lines. To ensure that our train and test splits are sufficiently distinct, we cluster perturbations based on their log-fold change and assign each cluster to the same split.

While perturbation target prediction appears to be a simple classification task, there are two aspects of the data that render standard metrics less meaningful. Not all genes can be mapped to the baselines' domain knowledge, so their effects as perturbations cannot be predicted. In addition, due to genetic redundancy, it is common for multiple perturbations to elicit similar responses (Kernfeld et al., 2023). Therefore, an "incorrectly" predicted target (e.g. out of 1000 genes) may not necessarily indicate poor performance. Therefore, we propose the following metrics.

- We report the **rank** of the ground truth target, normalized by the number of candidate genes. Here, 1 indicates that the ground truth target is at the top of the list (best), while 0 indicates that the ground truth is at the bottom (worst).

- To emulate a "virtual screening" setting, we report **recall at k** (the proportion of targets recovered within the top $k$ candidates). This value ranges from 0 to 1, where 1 is best.

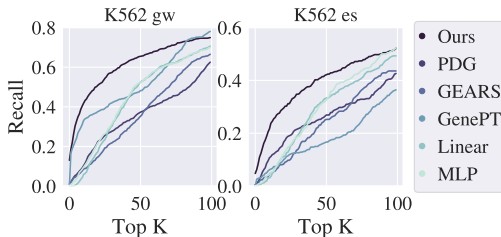

Figure 4: Recall at $k$ on K562 datasets.

Table 3: Rank of drug targets on chemical perturbation datasets (McFaline-Figueroa et al., 2024). A172 and T98G are *unseen* cell lines. For details and full drug names, please see Appendix B.1.

| Model | A172 | | | T98G | | |
|---|---|---|---|---|---|---|
| | infig. | nint. | palb. | doxo. | palb. | vola. |
| PDG | 0.719 | 0.346 | 0.395 | 0.055 | 0.345 | 0.420 |
| CDN | **0.883** | **0.436** | **0.582** | **0.059** | **0.543** | **0.647** |

- The **Pearson correlation** $r$ measures the similarity between the (ground truth) mean log-fold change of the top prediction that was observed as a perturbation, and that of the actual perturbation (given as input). These values range from -1 to 1, where 1 is best.

**Baselines** We compare to two classes of algorithms related to biological perturbations. Forward inference models predict the effects of perturbations, and perturbation targets can be ranked by comparing the inferred effect of each candidate to the interventional data.

- **GEARS** (Roohani et al., 2023) is a graph neural network that predicts the effects of unseen genetic perturbations, using the gene ontology knowledge graph (Ashburner et al., 2000) to model the relationship between genes. We trained GEARS on each cell line separately, and predicted the effects of perturbing every expressed gene. Then we ranked candidates based on cosine similarity to the interventional data.
- **GENEPT** (Chen & Zou, 2023) is a set of natural language-based embeddings that have been shown to achieve state-of-the-art performance on unseen perturbation effect prediction with simple downstream models (Märtens et al., 2024). We concatenated each perturbation's gene embedding to the log-fold change of the top 5000 highly-variable genes (Wolf et al., 2018) and trained a logistic regression model on each cell line to predict true vs. decoy perturbations. Candidates were ranked based on predicted probability.

Predictive models directly predict perturbation targets from data as a classification task.

- **LINEAR** and **MLP** take as input the mean expression of all perturbation targets, plus the top 2000 highly-variable genes (Wolf et al., 2018). They are trained to predict a binary label for each gene on each cell line independently.
- **PDGRAPHER** (Gonzalez et al., 2024) is a graph neural network that predicts the perturbation targets of genetic or chemical perturbations, on *seen* or *unseen* cell lines, with the human reference interactome (Luck et al., 2020) as the knowledge graph.

All baselines were run with their official implementations and/or latest releases. For more details, please see Appendix C.1.

**Results** Table 2 reports our results on five Perturb-seq datasets. In the majority of cases, CDN outperforms baselines, both in the seen and unseen cell line settings. This is more evident in Figure 4, in which CDN achieves higher recall at $k$ at nearly all points on the curves. While no baseline is as consistent as CDN, GENEPT is surprisingly competitive against more complex baselines. This may indicate that natural language induces embedding spaces with favorable geometries with respect to biological function. On the two chemical perturbation datasets, CDN ranks the ground truth targets higher than PDGRAPHER on all six drugs (Table 3). However, the mutually low performance on doxorubicin (doxo.) in T98G cells may indicate a failure mode of both models.

## 4.2 SYNTHETIC EXPERIMENTS

There are two considerations that motivate further experiments in controlled settings. First, while it would be ideal to evaluate all algorithms on real data, current causal discovery algorithms that support unknown interventions are not tractable on datasets with more than tens of variables. In fact, many baselines require hours on even $N = 10$ datasets, and they do not scale favorably (Table 9).

Table 2: Results on 5 Perturb-seq datasets. Top: "trivial" perturbations. Bottom: "non-trivial" perturbations. Primary setting: train all cell lines jointly, test on unseen perturbations. Suffix -CL: leave out the one cell line from training, test on the same set of unseen perturbations in the unseen cell line. Metrics, from left to right: normalized rank of ground truth, top 20 recall, and top 1 Pearson correlation. For ablations, runtimes, and uncertainties, see Appendix D.

| Model | K562 gw | | | K562 es | | | RPE1 | | | HepG2 | | | Jurkat | | |
|---|---|---|---|---|---|---|---|---|---|---|---|---|---|---|---|
| | rank | $R_{20}$ | $r$ | rank | $R_{20}$ | $r$ | rank | $R_{20}$ | $r$ | rank | $R_{20}$ | $r$ | rank | $R_{20}$ | $r$ |
| RANDOM | 0.50 | 0.13 | 0.16 | 0.50 | 0.09 | 0.17 | 0.50 | 0.09 | 0.36 | 0.50 | 0.06 | 0.34 | 0.50 | 0.08 | 0.22 |
| LINEAR | 0.49 | 0.16 | 0.21 | 0.50 | 0.13 | 0.31 | 0.48 | 0.09 | 0.45 | 0.48 | 0.07 | 0.43 | 0.48 | 0.08 | 0.29 |
| MLP | 0.48 | 0.21 | 0.20 | 0.47 | 0.13 | 0.21 | 0.54 | 0.19 | 0.37 | 0.51 | 0.10 | 0.42 | 0.48 | 0.13 | 0.32 |
| GENEPT | 0.60 | 0.39 | 0.39 | 0.47 | 0.11 | 0.26 | 0.57 | 0.33 | 0.55 | 0.44 | 0.12 | 0.41 | 0.55 | **0.32** | **0.53** |
| GEARS | 0.49 | 0.14 | 0.18 | 0.51 | 0.09 | 0.27 | 0.49 | 0.09 | 0.40 | 0.48 | 0.09 | 0.40 | 0.47 | 0.06 | 0.24 |
| PDG | 0.49 | 0.20 | 0.26 | 0.52 | 0.18 | 0.34 | 0.51 | 0.11 | 0.42 | 0.46 | 0.06 | 0.40 | 0.49 | 0.11 | 0.35 |
| CDN | **0.77** | **0.52** | **0.49** | **0.68** | **0.32** | **0.45** | **0.74** | **0.37** | **0.55** | **0.68** | **0.20** | **0.50** | **0.67** | 0.27 | 0.45 |
| PDG-CL | 0.40 | 0.04 | 0.15 | 0.41 | 0.02 | 0.24 | 0.48 | 0.02 | 0.37 | 0.47 | 0.03 | 0.41 | 0.53 | 0.06 | 0.33 |
| CDN-CL | **0.71** | **0.43** | **0.40** | **0.66** | **0.27** | **0.42** | **0.62** | **0.24** | **0.53** | **0.66** | **0.17** | **0.46** | **0.65** | **0.26** | **0.47** |
| RANDOM | 0.50 | 0.11 | 0.17 | 0.50 | 0.04 | 0.17 | 0.50 | 0.04 | 0.44 | 0.50 | 0.04 | 0.33 | 0.50 | 0.06 | 0.21 |
| LINEAR | 0.51 | 0.18 | 0.23 | 0.48 | 0.01 | 0.25 | 0.49 | 0.03 | 0.52 | 0.51 | 0.04 | **0.43** | 0.50 | 0.06 | 0.23 |
| MLP | 0.45 | 0.19 | 0.21 | 0.49 | 0.04 | 0.11 | 0.53 | 0.09 | 0.41 | 0.48 | **0.07** | 0.40 | 0.45 | 0.07 | 0.29 |
| GENEPT | 0.52 | 0.24 | 0.32 | 0.42 | 0.03 | 0.19 | 0.41 | 0.06 | 0.46 | 0.31 | 0.03 | 0.40 | 0.51 | **0.15** | 0.36 |
| GEARS | 0.56 | 0.16 | 0.22 | 0.45 | 0.06 | 0.20 | 0.49 | 0.04 | 0.44 | 0.54 | 0.03 | 0.40 | 0.50 | 0.11 | 0.26 |
| PDG | 0.54 | 0.24 | 0.29 | 0.49 | 0.03 | 0.23 | 0.48 | 0.05 | 0.52 | 0.54 | 0.05 | **0.43** | 0.52 | 0.10 | 0.35 |
| CDN | **0.72** | **0.37** | **0.32** | **0.56** | **0.15** | **0.35** | **0.59** | **0.12** | **0.53** | **0.55** | **0.07** | 0.40 | **0.62** | 0.10 | **0.39** |
| PDG-CL | 0.39 | 0.02 | 0.13 | 0.44 | 0.01 | **0.28** | **0.54** | 0.02 | **0.48** | 0.48 | 0.02 | **0.41** | 0.57 | 0.06 | 0.35 |
| CDN-CL | **0.68** | **0.25** | **0.29** | **0.55** | **0.06** | 0.27 | 0.49 | **0.07** | **0.48** | **0.59** | **0.05** | 0.39 | **0.60** | **0.18** | **0.38** |

Our transcriptomics datasets contain hundreds of genes, even after filtering to those that are differentially expressed (Figures 6 and 7). On the other hand, existing models for biological perturbations all rely on some form of domain knowledge, and their performance is inseparable from the choice and quality of these external data. Here, synthetic settings allow us to assess the model's capacity to predict intervention targets in isolation.

**Datasets** We generated 120 synthetic test datasets with hard and soft interventions. To generate observational data, we sampled Erdős-Rényi graphs with $N = 10, 20$ nodes and $E = N, 2N$ expected edges; causal mechanism parameters; and observations of each variable, in topological order. We sampled $3N$ distinct subsets of 1-3 nodes ($N$ each) as intervention targets. For hard interventions, we set $x \leftarrow z$, where $z$ is uniform. To emulate transcriptomics data, in which perturbation effects are measured in fold change, we introduce soft interventions: $x \leftarrow cf(\pi_x)$, where $c$ is a positive scaling factor, with equal probability $c \lessgtr 1$. For our synthetic training set, we generated approximately 4000 datasets following the training set of SEA, with hard interventions only. Full details are available in Appendix B.2.

**Baselines** We compare against discrete and continuous causal discovery algorithms for unknown interventions. **UT-IGSP** (Squires et al., 2020) infers causal graphs and unknown targets by greedily selecting the permutation of variables that minimizes their proposed score function. **DCDI** (Brouillard et al., 2020) and **BACADI** (Hägele et al., 2023) are continuous causal discovery algorithms that fit generative models to the data, where the causal graph and intervention targets are model parameters. The -G and -DSF suffixes on DCDI correspond to Gaussian and deep sigmoidal flow parametrizations of the likelihood. BACADI is evaluated here using the fully-connected implementation (better performance), with the linear version in Table 11. Its -E and -M suffixes indicate empirical (standard) and mixture (bootstrap) variants.

We also compare against the difference causal inference (**DCI**) algorithm with stability selection (Belyaeva et al., 2021), a discrete optimization algorithm that aims to detect edge-level differences between two causal graphs. On this node-centric task, we take each node's proportion of changed edges as its likelihood of being an intervention target. While DCI was also motivated by

Table 4: Intervention target prediction results on synthetic datasets with $N = 10, E = 10$. Uncertainty is standard deviation over 5 distinct datasets. Top: hard interventions; bottom: soft interventions ("scale"). Number in parentheses indicates number of intervention targets. Runtimes are documented in Table 9, and full results are available in Table 11.

| Type | Model | Linear (1) | | Linear (3) | | Polynomial (1) | | Polynomial (3) | |
|---|---|---|---|---|---|---|---|---|---|
| | | mAP↑ | AUC↑ | mAP↑ | AUC↑ | mAP↑ | AUC↑ | mAP↑ | AUC↑ |
| Hard | UT-IGSP | $0.19_{\pm.03}$ | $0.56_{\pm.05}$ | $0.38_{\pm.04}$ | $0.57_{\pm.05}$ | $0.27_{\pm.05}$ | $0.63_{\pm.03}$ | $0.38_{\pm.03}$ | $0.55_{\pm.03}$ |
| | DCDI-G | $0.38_{\pm.12}$ | $0.59_{\pm.11}$ | $0.47_{\pm.03}$ | $0.53_{\pm.03}$ | $0.27_{\pm.09}$ | $0.47_{\pm.07}$ | $0.45_{\pm.04}$ | $0.49_{\pm.05}$ |
| | DCDI-DSF | $0.32_{\pm.05}$ | $0.58_{\pm.10}$ | $0.43_{\pm.03}$ | $0.48_{\pm.03}$ | $0.25_{\pm.04}$ | $0.42_{\pm.05}$ | $0.46_{\pm.03}$ | $0.50_{\pm.05}$ |
| | DCI | $0.42_{\pm.07}$ | $0.75_{\pm.03}$ | $0.63_{\pm.11}$ | $0.80_{\pm.08}$ | $0.39_{\pm.10}$ | $0.72_{\pm.06}$ | $0.53_{\pm.09}$ | $0.69_{\pm.07}$ |
| | BACADI-E | $0.16_{\pm.04}$ | $0.61_{\pm.09}$ | $0.36_{\pm.04}$ | $0.60_{\pm.06}$ | $0.13_{\pm.02}$ | $0.62_{\pm.05}$ | $0.36_{\pm.03}$ | $0.60_{\pm.04}$ |
| | BACADI-M | $0.12_{\pm.01}$ | $0.57_{\pm.05}$ | $0.33_{\pm.02}$ | $0.56_{\pm.04}$ | $0.12_{\pm.01}$ | $0.57_{\pm.04}$ | $0.33_{\pm.02}$ | $0.55_{\pm.04}$ |
| | CDN (MLP) | $0.72_{\pm.06}$ | $\mathbf{0.88}_{\pm.04}$ | $\mathbf{0.87}_{\pm.04}$ | $\mathbf{0.91}_{\pm.04}$ | $0.56_{\pm.09}$ | $0.78_{\pm.09}$ | $0.73_{\pm.08}$ | $\mathbf{0.80}_{\pm.06}$ |
| | CDN (AXIAL) | $\mathbf{0.73}_{\pm.08}$ | $\mathbf{0.88}_{\pm.04}$ | $0.83_{\pm.06}$ | $0.88_{\pm.05}$ | $\mathbf{0.60}_{\pm.13}$ | $\mathbf{0.80}_{\pm.10}$ | $\mathbf{0.74}_{\pm.07}$ | $\mathbf{0.80}_{\pm.06}$ |
| Soft | UT-IGSP | $0.20_{\pm.02}$ | $0.71_{\pm.04}$ | $0.37_{\pm.01}$ | $0.54_{\pm.03}$ | $0.24_{\pm.03}$ | $0.72_{\pm.03}$ | $0.39_{\pm.04}$ | $0.58_{\pm.05}$ |
| | DCDI-G | $0.38_{\pm.11}$ | $0.58_{\pm.11}$ | $0.48_{\pm.05}$ | $0.50_{\pm.07}$ | $0.25_{\pm.06}$ | $0.49_{\pm.08}$ | $0.48_{\pm.05}$ | $0.53_{\pm.06}$ |
| | DCDI-DSF | $0.37_{\pm.09}$ | $0.59_{\pm.11}$ | $0.45_{\pm.05}$ | $0.51_{\pm.06}$ | $0.32_{\pm.09}$ | $0.53_{\pm.13}$ | $0.43_{\pm.04}$ | $0.47_{\pm.04}$ |
| | DCI | $0.45_{\pm.10}$ | $\mathbf{0.73}_{\pm.07}$ | $0.54_{\pm.04}$ | $\mathbf{0.68}_{\pm.03}$ | $0.34_{\pm.03}$ | $0.69_{\pm.03}$ | $0.45_{\pm.03}$ | $0.62_{\pm.01}$ |
| | BACADI-E | $0.34_{\pm.16}$ | $0.63_{\pm.10}$ | $0.49_{\pm.14}$ | $0.62_{\pm.15}$ | $0.22_{\pm.03}$ | $0.64_{\pm.05}$ | $0.38_{\pm.06}$ | $0.58_{\pm.06}$ |
| | BACADI-M | $0.28_{\pm.11}$ | $0.62_{\pm.10}$ | $0.44_{\pm.14}$ | $0.59_{\pm.15}$ | $0.21_{\pm.02}$ | $0.64_{\pm.05}$ | $0.38_{\pm.06}$ | $0.58_{\pm.06}$ |
| | CDN (MLP) | $0.29_{\pm.04}$ | $0.53_{\pm.05}$ | $0.44_{\pm.04}$ | $0.47_{\pm.02}$ | $0.33_{\pm.09}$ | $0.50_{\pm.09}$ | $0.52_{\pm.06}$ | $0.56_{\pm.05}$ |
| | CDN (AXIAL) | $\mathbf{0.46}_{\pm.11}$ | $0.68_{\pm.08}$ | $\mathbf{0.58}_{\pm.09}$ | $0.62_{\pm.11}$ | $\mathbf{0.81}_{\pm.09}$ | $\mathbf{0.92}_{\pm.07}$ | $\mathbf{0.75}_{\pm.05}$ | $\mathbf{0.77}_{\pm.06}$ |

biological applications, it only scales to around a hundred nodes at most, so we evaluate it alongside other causal discovery methods here.

**Evaluation**  In the synthetic case, we are not constrained by biological redundancy or incomplete predictions, so we report standard classification metrics.

- **Mean average precision** (mAP) is the average precision over a continuum of binarization thresholds, computed independently for each variable, averaged over all regimes of the same number of targets. This ranges from 0 to 1 (perfect), where the random baseline depends on the positive rate.

- **Area under the receiver operating curve** (AUC) is also computed independently for each variable, averaged over the same set. This ranges from 0 to 1 (perfect), where the random baseline is 0.5 (per edge).

**Results**  On synthetic data, CDN achieves high performance across intervention types and data-generating mechanisms (Table 4), while running in seconds (Table 9). As an ablation study, we investigated replacing the differential network with a multi-layer perceptron, which does not model interactions between edges. On the hard interventions, the MLP is sufficient for predicting intervention targets, perhaps by abusing marginal variances (Reisach et al., 2021) or other node-level artifacts of synthetic data. Once we consider soft interventions, however, the original CDN formulation significantly outperforms the MLP variant – indicating that graph-level information is essential.

DCI is the best baseline, which is encouraging, as it was designed for the edge-centric version of our task, with the intuition that detecting edge differences is easier than reproducing the entire causal graph. It performs particularly well on linear data, which align with its assumptions, and less well on non-linear data. Surprisingly, most other baselines perform poorly at recovering intervention targets. In the case of DCDI and BACADI, this may be because it is hard to select a single sparsity threshold for varying sizes of intervention sets, and to balance the regularizer with the generative modeling objective.

## 5 CONCLUSION

We presented CDN, a causal discovery-powered approach for predicting perturbation targets in the context of single-cell transcriptomics. CDN uses an amortized causal discovery algorithm to represent a pair of observational and interventional datasets in terms of their data-generating mechanisms, and then trains a classifier to identify variables whose conditional independencies have changed. Our approach achieves the state-of-the-art in predicting perturbation targets over seven transcriptomics datasets, compared to a variety of perturbation modeling baselines. CDN also surpasses causal discovery algorithms for predicting unknown targets across synthetic settings, while only requiring a fraction of the runtime. Nonetheless, there is substantial space for improvement, so we hope that this work and the associated datasets will inform future method development for these biological applications.

## REPRODUCIBILITY STATEMENT

All source code is available in the supplementary materials. Hyperparameter and implementation choices are described in Section 3.3. Details regarding biological data pre-processing can be found in Section 4.1 and Appendix B.1. The raw data are freely available via the accession codes listed in Table 5. Synthetic data generation is discussed in Appendix B.2. Details required to reproduce baselines are described in Sections 4.1 and 4.2, as well as Appendix C.1.

## ETHICS STATEMENT

We use publicly available datasets which do not involve human subjects. While our work has applications to biological understanding, our work does not directly concern the design of molecules or other potentially harmful agents.

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

# A THEORETICAL INTUITION

While this paper focuses on biological applications, we would like to show that our model is well-specified, i.e. it has the capacity to predict correct intervention targets, given certain inputs.

**Preliminary note**   Continuous causal discovery works typically rely on the universality of their architectures for consistency (Brouillard et al., 2020), so this analysis also focuses on computational capacity. Theoretically, it is impossible to "guarantee" that high-dimensional representations from learned model contain or do not contain certain information. This can be tested empirically through probing, e.g. querying for grammatical structure in language models (Vulić et al., 2020), but cannot be "proven" in the traditional sense. Thus, we believe that the empirical performance of our model on real datasets is its primary contribution, but provide these explanations for interested readers. In this analysis, we assume that the "causal" representations $h$ contains enough information both to retain global statistics and recover the true graph $E$. The former is reasonable due to high model capacity, and the latter is based on high empirical performance in graph reconstruction (Wu et al., 2024). We emphasize that the latter is an empirical judgment, which may not hold on all datasets in practice.

**Axial attention architecture**   Our differential network is implemented using an axial attention layer, which is composed of two self-attention layers (one along each axis of the adjacency matrix) and a feed-forward network. For simplicity, we follow prior work Yun et al. (2019) and ignore layer normalization and dropout.

Our inputs $h \in \mathbb{R}^{2d \times N \times N}$ are $2d$-dimension features, which represent a pair of $N \times N$ causal graphs. We use $h_{\cdot,j}$ to denote a length $N$ row for a fixed column $j$, and $h_{i,\cdot}$ to denote a length $N$ column for a fixed row $i$. The axial attention layer implements:

$$\text{Attn}_{\text{row}}(h_{\cdot,j}) = h_{\cdot,j} + W_O W_V h_{\cdot,j} \cdot \sigma \left[ (W_K h_{\cdot,j})^T W_Q h_{\cdot,j} \right],$$

$$\text{Attn}_{\text{col}}(h_{i,\cdot}) = h_{i,\cdot} + W_O W_V h_{i,\cdot} \cdot \sigma \left[ (W_K h_{i,\cdot})^T W_Q h_{i,\cdot} \right],$$

$$\text{FFN}(h) = h + W_2 \cdot \text{ReLU}(W_1 \cdot h + b_1) + b_2,$$

where $W_O \in \mathbb{R}^{2d \times 2d}, W_V, W_K, W_Q \in \mathbb{R}^{2d \times 2d}, W_2 \in \mathbb{R}^{2d \times m}, W_1 \in \mathbb{R}^{m \times 2d}, b_2 \in \mathbb{R}^{2d}, b_1 \in \mathbb{R}^m$, and $m$ is the FFN hidden dimension. We have omitted the $i$ and $j$ subscripts on the $W$s, but they use separate parameters. Any self attention can take on the identity mapping by setting $W_O, W_V, W_K, W_Q$ to $2d \times 2d$ matrices of zeros.

**Hard interventions**   Let $G = (V, E)$ be a causal graphical model associated with data distribution $P_X$. Let $G' = (V, E')$ and $\tilde{P}_X$ denote the causal graph and data distribution after an unknown intervention, with ground truth targets $I \subsetneq V$. For convenience, we use $E, E'$ both to denote sets of edges, as well as the equivalent adjacency matrices.

In the case of perfect interventions,

$$f(x_i) \leftarrow z_i, \forall i \in I \tag{4}$$

where $z_i \perp\!\!\!\perp X$ are independent random variables. $\tilde{P}_X$ is associated with mutilated graph $E'$, where

$$E' = E \setminus \bigcup_{i \in I} \{(j,i)\}_{(j,i) \in E}. \tag{5}$$

In terms of the associated adjacency matrices, $E - E'$ has 1s in each column $i \in I$ and 0s elsewhere.

Here, the axial attention layer should implement $h - h'$, so that when we collapse over the incoming edges, the output is non-zero only at the edge differences. Suppose the first dimension of the $2d$ feature stores $E$, and the second dimension stores $E'$. The row self-attention implements the identity (in the first two dimensions). Then we can set $W_{K,Q}$ to zero, $W_V$ to the identity, and $W_O$ to

$$W_O = \begin{bmatrix} 1 & -1 \\ 0 & 0 \end{bmatrix} - \begin{bmatrix} 1 & 0 \\ 0 & 1 \end{bmatrix} \tag{6}$$

to account for the residual. The FFN implements the identity, so that when we take the mean over along the rows, we recover non-zero elements at all nodes whose incoming edges were removed.

**Soft interventions**  We also study soft interventions the context of causal models with non-multiplicative noise, in which intervention targets are scaled by constant factors,

$$f(x_i) \leftarrow c_i f(x), c_i > 0 \tag{7}$$

where $c_i$ are sampled at random per synthetic dataset. This choice is inspired by the fact that biological perturbation effects are measured in fold-change. Here, the adjacency matrices are the same, but global statistics differ. In particular, we focus on two statistics: the correlation matrix $R$ and the covariance matrix $\Sigma$. Note that while these inputs differ slightly from our main experiments (reasons discussed in Appendix D), we show that they still achieve reasonable performance in Table 8.

Suppose $x$ is an intervention target.

- $R - R'$ is non-zero in all entries $i, j$ and $j, i$ where $i$ is a descendent of $x$, and $j$ is any node for which $R_{i,j} \neq 0$ (e.g. ancestors, descendants, and $x$, if $P_X$ is faithful to $G$).

- $\Sigma - \Sigma'$ is non-zero in all entries $i, j$ and $j, i$ where $i$ is a descendent of $x$ or $i = x$, and $j$ is any node for which $\Sigma_{i,j} \neq 0$ (e.g. ancestors, descendants, and $x$, if $P_X$ is faithful to $G$).

All descendants are always affected, due to the non-multiplicative noise term. These two differ in the row and column that correspond to $x$ since

$$\text{Corr}(c \cdot x, y) = \text{Corr}(x, y) \tag{8}$$
$$\text{Cov}(c \cdot x, y) = c \cdot \text{Cov}(x, y). \tag{9}$$

Therefore, to identify $x$, we should find the index in which $\Sigma$ differs but not $R$. Suppose that dimensions 3-6 of $h$ encode $R, R', \Sigma, \Sigma'$. Following the same strategy as the hard interventions, we can use the row attention to compute $R - R', \Sigma - \Sigma'$ and store them in dimensions 3, 4. Then we use the column attention to filter out variables that are independent from $x$ by storing the sum of each column in dimensions 5, 6. While not strictly impossible, it is unlikely that a variable dependent on $x$ would result in a column that sums to exactly 0. Thus, all columns with non-zero sums are either ancestors, descendants, or $x$. The feedforward network implements

$$\text{FFN}(h_{\cdot,3-6}) = \begin{cases} 1 & h_{\cdot,3} = 0, h_{\cdot,4} \neq 0, h_{\cdot,5} \neq 0 \\ 0 & \text{otherwise.} \end{cases} \tag{10}$$

This results in 1s in the rows and columns where $\Delta R$ and $\Delta \Sigma$ differ. After collapsing over incoming edges and normalizing to probabilities, the maximum probabilities can be found at the intervention targets.

**Supporting both intervention types**  Recall that the final output layer is a linear projection from $2d$ to 1. If this layer implements a simple summation over all $2d$, the predicted intervention targets are consistent with both hard and soft interventions. For soft interventions, $E = E'$, so the hard intervention dimensions will be 0. Likewise, for hard interventions, both $R$ and $\Sigma$ will differ as the same locations, as the underlying variable has changed, so the soft intervention dimensions will be 0. Since the two techniques produce mutually exclusive predictions, this means that both hard and soft interventions can co-exist and be detected on different nodes.

Finally, we re-iterate that CDN operates over high-dimensional features, rather than the inputs and outputs themselves. Thus, these sketches only serve to provide context regarding the axial attention framework's computational capacity, rather than as a blueprint for the computations it performs.

## B  DATASETS

### B.1  BIOLOGICAL DATASETS

We converted all single cell datasets to LogTP10K + 1 expression values (log-normalized, transcripts per 10,000 UMIs). Perturb-seq dataset variables represented genes that were mapped and filtered by the authors. Sci-Plex dataset variables represented genes that appeared in at least 5,000 cells (threshold chosen to achieve a similar number of genes). We performed differential expression analysis via the `scanpy` package (Wolf et al., 2018), using the Wilcoxon signed-rank test (Wilcoxon, 1945)

Table 5: Extended biological dataset statistics (raw).

| Type | Source | Accession | Cell line | # Perts | # Genes | # NTCs | # Cells |
|------|--------|-----------|-----------|---------|---------|--------|---------|
| Genetic | Replogle et al. (2022) | Figshare 20029387 | K562 gw | 9,866 | 8,248 | 75,328 | 1,989,578 |
| | | | K562 es | 2,057 | 8,563 | 10,691 | 310,385 |
| | | | RPE1 | 2,393 | 8,749 | 11,485 | 247,914 |
| | Nadig et al. (2024) | GSE220095 | HepG2 | 2,393 | 9,624 | 4,976 | 145,473 |
| | | | Jurkat | 2,393 | 8,882 | 12,013 | 262,956 |
| Chemical | McFaline-Figueroa et al. (2024) | GSM7056151 | A172 | 23 | 8,393 | 8,660 | 58,347 |
| | | | T98G | 23 | 8,393 | 6,921 | 58,347 |

Table 6: Extended biological dataset statistics (processed).

| | Perturbations | | | | Genes | | Cells |
|---------|-------|------|---------|-------------|----------|------------|--------|
| Dataset | Train | Test | Trivial | Non-trivial | Unique # DE | Median # DE | |
| K562 gw | 1089 | 678 | 587 | 91 | 7,378 | 81 | 492,096 |
| K562 es | 640 | 420 | 348 | 72 | 8,492 | 226 | 213,552 |
| RPE1 | 564 | 397 | 233 | 164 | 8,641 | 399 | 179,696 |
| HepG2 | 364 | 263 | 162 | 101 | 9,282 | 271 | 79,309 |
| Jurkat | 679 | 333 | 262 | 71 | 8,432 | 162 | 174,698 |
| A172 | — | 3 | — | — | 445 | 324 | 18,196 |
| T98G | — | 3 | — | — | 1,644 | 508 | 13,126 |

with Benjamini-Hochberg p-value correction and a threshold of adjusted p-value $< 0.05$. Table 5 reports statistics of the raw, unprocessed datasets.

For Perturb-seq datasets, we kept perturbations with $> 10$ differentially-expressed genes (DEGs), and clustered them using k-means with $k = 200$, chosen heuristically based on log-fold change heatmaps (Figure 5). These clusters were used to inform data splits for seen cell lines, where the largest cluster was allocated to the training set, and all remaining clusters were split equally among train and test. The largest cluster(s) appear to contain perturbations with smaller effects. For Sci-Plex datasets, only 6 drug perturbations across 2 cell lines resulted in differential expression of their known protein targets (Supplementary Table 8 from McFaline-Figueroa et al. (2024)). Therefore, we used these exclusively as test sets. Table 6 reports statistics of the final, processed datasets. Figures 6 and 7 plot the full distribution of number of cells and DEGs per perturbation.

The full names of the Sci-Plex drugs are as follows.

- Infigratinib ("infig")

- Nintedanib ("nint")

- Palbociclib ("palb")

- Doxorubicin ("doxo")

- Volasertib ("vola")

## B.2 SYNTHETIC DATA

Synthetic data were generated using code modified from DCDI (Brouillard et al., 2020) for soft interventions. We used the following implementations for causal mechanisms $f(x)$, where $x$ is the variable in question, $M$ is a binary mask for the parents of $x$, $X$ contains measurements of all variables, $E$ is independent Gaussian noise, and $W$ is a random weight matrix.

- Linear: $f(x) = MXW + E$.

- Polynomial: $f(x) = W_0 + MXW_1 + MX^2W_2 + E$

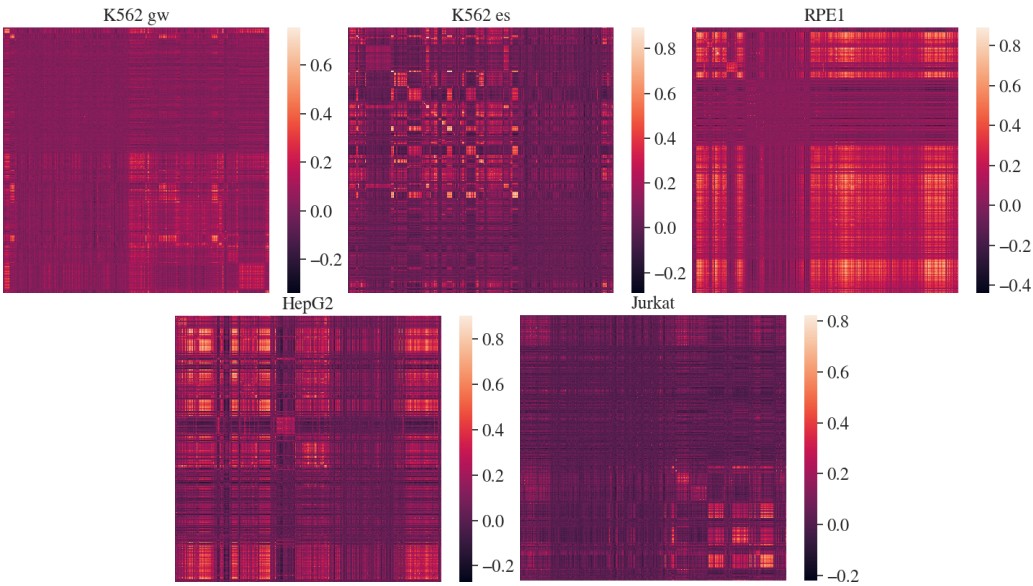

Figure 5: Heatmap of correlation between log-fold change, sorted by cluster.

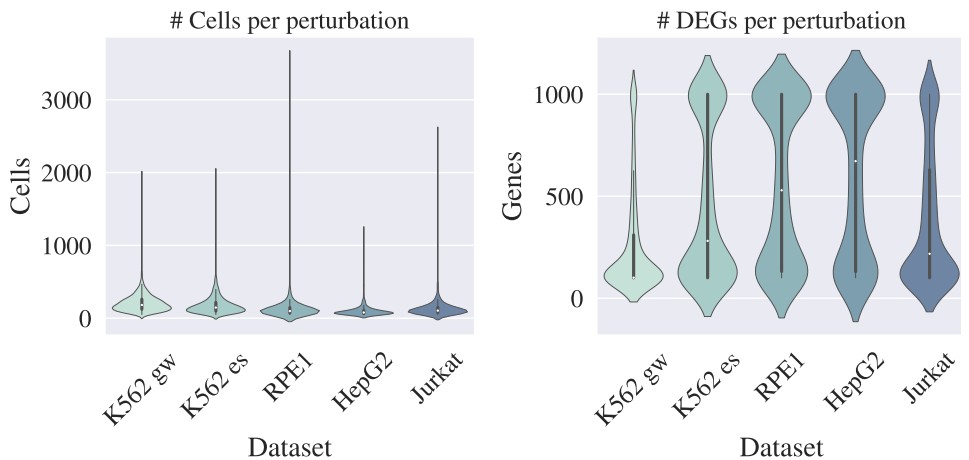

Figure 6: Perturb-seq dataset statistics, after processing (Table 6).

Root causal mechanisms are uniform. For hard interventions, we set $f(x) \leftarrow z$, where $z \sim$ Uniform$(-1, 1)$. For soft interventions, we set

$$f(x) \leftarrow z_1^{\text{Sign}(z)} \cdot f(x)$$
$$z_1 \sim \text{Uniform}(2, 4)$$
$$z \sim \text{Uniform}(-1, 1).$$

That is, we multiply $f(x)$ by a scaling factor that is equal probability $\lessgtr 1$ (constant across all observations).

## C  IMPLEMENTATION DETAILS

### C.1  BASELINES

We used the latest releases of all baselines. For GENEPT, this corresponds to the v2 March 2024 update, which used newer models and additional protein data, compared to their initial paper (`GenePT_gene_protein_embedding_model_3_text`).

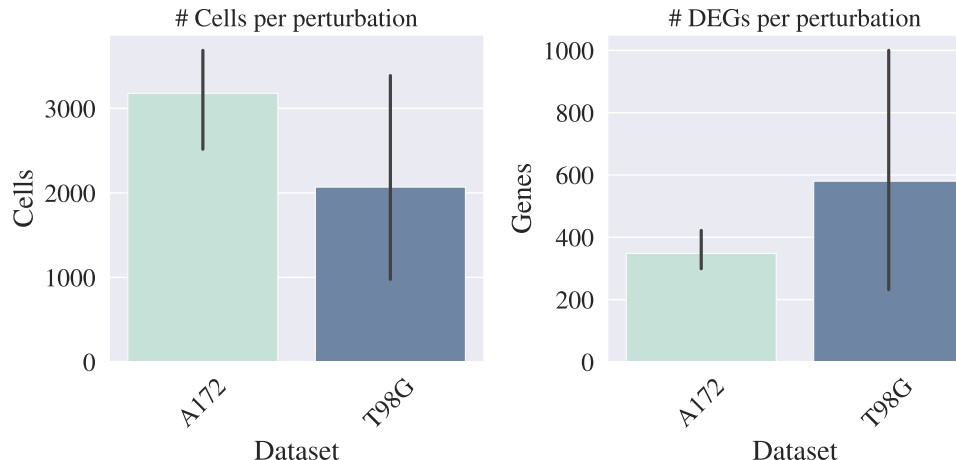

Figure 7: Sci-Plex dataset statistics, after processing. There are 3 chemical perturbations per cell line.

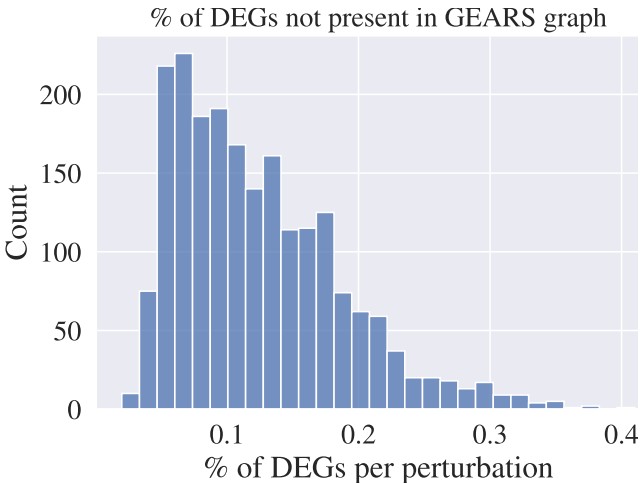

Figure 8: Percentage of differentially expression genes (per perturbation) that were *not* in the GEARS knowledge graph, and whose effects could not be predicted.

Due to the varying coverage of genes in their domain knowledge, several baselines failed to make predictions for certain genes. Of the 2,842 unique genetic perturbations, only 2,819 (99.1%) mapped to GENEPT embeddings. The remainder used the mean gene embedding (within the dataset) as the language-based embedding.

Figure 8 depicts the low, but non-trivial proportion of differentially expressed genes for which GEARS was unable to make a prediction, due to lack of node coverage in their processed gene ontology graph (Ashburner et al., 2000). These genes were not considered in the rankings or evaluations for GEARS.

PDGRAPHER was published on the union of three distinct knowledge graphs, but their harmonized graphs were not available publicly, and the authors did not respond to requests for data sharing. As a result, we relied on solely the human reference interactome (Luck et al., 2020) graph, as it was the only one of the three that could be easily processed. All test perturbation targets could be inferred through PDGRAPHER.

Table 7: Ablation studies. We re-train CDN without pretrained aggregator weights, marginal estimates, or global statistics.

| Model | K562 gw | | | K562 es | | | RPE1 | | | HepG2 | | | Jurkat | | |
|---|---|---|---|---|---|---|---|---|---|---|---|---|---|---|---|
| | rank | $R_{20}$ | $r$ | rank | $R_{20}$ | $r$ | rank | $R_{20}$ | $r$ | rank | $R_{20}$ | $r$ | rank | $R_{20}$ | $r$ |
| CDN | **0.72** | **0.37** | **0.32** | **0.56** | **0.15** | **0.35** | **0.59** | **0.12** | **0.53** | **0.55** | **0.07** | **0.40** | **0.62** | **0.10** | **0.39** |
| −pretrain | 0.50 | 0.14 | 0.20 | 0.51 | 0.01 | 0.23 | 0.49 | 0.03 | 0.50 | 0.52 | 0.03 | 0.37 | 0.54 | 0.07 | 0.30 |
| −marginal | 0.55 | 0.15 | 0.19 | 0.49 | 0.06 | 0.25 | 0.50 | 0.05 | 0.47 | 0.49 | 0.05 | 0.39 | 0.58 | 0.04 | 0.31 |
| −global | 0.52 | 0.14 | 0.20 | 0.49 | 0.04 | 0.27 | 0.47 | 0.04 | 0.49 | 0.49 | 0.04 | 0.35 | 0.46 | 0.06 | 0.25 |

Table 8: Ablation study on global statistics over synthetic datasets. MLP and AXIAL use inverse covariance (same as pretraining). CORR was finetuned for correlation, and CORR+COV was finetuned for correlation *and* covariance (concatenated).

| $N$ | $E$ | CDN Variant | Linear (Hard) | | Linear (Soft) | | Poly. (Hard) | | Poly. (Soft) | |
|---|---|---|---|---|---|---|---|---|---|---|
| | | | mAP↑ | AUC↑ | mAP↑ | AUC↑ | mAP↑ | AUC↑ | mAP↑ | AUC↑ |
| 10 | 10 | MLP | **0.73**±.04 | **0.86**±.04 | 0.33±.03 | 0.50±.02 | 0.61±.08 | **0.77**±.07 | 0.39±.03 | 0.51±.04 |
| | | AXIAL | **0.73**±.05 | 0.85±.03 | **0.48**±.06 | **0.64**±.05 | **0.62**±.09 | 0.77±.07 | **0.73**±.07 | **0.82**±.06 |
| | | +CORR | 0.71±.07 | 0.84±.04 | 0.32±.05 | 0.47±.07 | 0.58±.08 | 0.74±.05 | 0.27±.04 | 0.34±.04 |
| | | +CORR+COV | 0.68±.04 | 0.82±.02 | 0.39±.03 | 0.50±.03 | 0.58±.07 | 0.71±.07 | 0.48±.05 | 0.65±.05 |
| 10 | 20 | MLP | **0.78**±.06 | **0.88**±.03 | 0.38±.05 | 0.56±.05 | **0.80**±.06 | **0.89**±.03 | 0.38±.05 | 0.48±.05 |
| | | AXIAL | 0.74±.05 | 0.86±.03 | **0.44**±.08 | **0.60**±.06 | 0.77±.05 | **0.88**±.03 | **0.54**±.05 | 0.65±.06 |
| | | +CORR | **0.78**±.03 | **0.88**±.02 | 0.35±.02 | 0.52±.02 | 0.73±.06 | 0.84±.03 | 0.34±.05 | 0.41±.06 |
| | | +CORR+COV | 0.75±.03 | 0.84±.02 | 0.34±.03 | 0.47±.06 | 0.69±.07 | 0.79±.05 | 0.53±.04 | **0.68**±.04 |
| 20 | 20 | MLP | **0.67**±.05 | **0.87**±.02 | 0.28±.01 | 0.54±.03 | **0.60**±.06 | **0.84**±.03 | 0.34±.04 | 0.51±.04 |
| | | AXIAL | 0.65±.03 | 0.86±.01 | **0.49**±.05 | **0.72**±.03 | 0.58±.05 | 0.83±.02 | **0.65**±.02 | **0.81**±.02 |
| | | +CORR | 0.62±.03 | 0.83±.01 | 0.25±.05 | 0.47±.05 | 0.58±.05 | 0.81±.02 | 0.21±.02 | 0.37±.03 |
| | | +CORR+COV | 0.60±.03 | 0.81±.01 | 0.31±.03 | 0.58±.01 | 0.58±.04 | 0.80±.02 | 0.40±.03 | 0.65±.03 |
| 20 | 40 | MLP | **0.78**±.04 | **0.92**±.01 | 0.29±.03 | 0.56±.03 | **0.72**±.07 | **0.88**±.04 | 0.36±.05 | 0.58±.04 |
| | | AXIAL | 0.74±.05 | 0.90±.02 | **0.38**±.03 | **0.65**±.01 | 0.68±.08 | 0.87±.04 | **0.64**±.03 | **0.79**±.03 |
| | | +CORR | 0.77±.07 | **0.92**±.03 | 0.29±.02 | 0.53±.02 | 0.63±.07 | 0.83±.04 | 0.23±.03 | 0.42±.02 |
| | | +CORR+COV | 0.73±.05 | 0.87±.02 | 0.36±.03 | 0.58±.03 | 0.62±.07 | 0.82±.03 | 0.43±.04 | 0.67±.02 |

# D  ADDITIONAL ANALYSES

**Ablation studies**  We run ablation studies on the model architecture (Table 7) by re-training CDN without pretrained aggregator weights, global statistics, or marginal estimates. We find that pretrained weights and global correlation are essential to model performance. The marginal estimates are of mixed importance: they are beneficial on the genome-wide (gw) dataset and Jurkat, but less on the others.

Table 8 ablates variants of CDN trained with different global statistics (but initialized with the inverse covariance pretrained weights) on synthetic data. The default summary statistic (MLP, AXIAL) is inverse covariance. CORR denotes the correlation version, later finetuned on Perturb-seq data (since correlation is much easier to compute on large graphs, compared to inverse covariance). The CORR+COV version aligns exactly with our theoretical intuition, and performs well in most settings (polynomial soft is the exception). This model modified the global layer to take an input of size $N \times N \times 2$, where each embedding was initialized to a distinct copy of the original. However, the inverse covariance version still performed the best, perhaps since there is no mismatch between the pretraining and finetuning statistics.

**Runtime**  We also compared the runtimes of various algorithms on the Perturb-seq and synthetic datasets. On the Perturb-seq data, all models finished running within minutes with the exception of GEARS (Figure 9). Runtimes of the various causal algorithms varied significantly (Table 9). The slowest method was DCDI, with an average runtime of around 10 hours on $N = 20$ datasets, while the fastest were UT-IGSP and the MLP variant of CDN. All models were benchmarked on equivalent hardware (A6000 GPU, 1 CPU core).

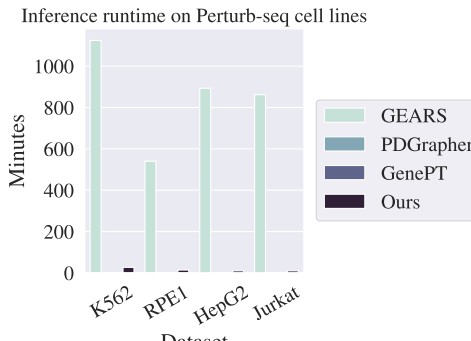

Figure 9: Inference runtimes on Perturb-seq datasets. K562 datasets are reported together. All models run on a single A6000 GPU, no constraint on memory (up to 500G). Only GEARS required over ~20G of memory. To the best of our ability, we normalized batch size to 1.

Table 9: Runtimes on synthetic datasets (sec).

| Nodes | Model | Min | Max | Mean | Std |
|---|---|---|---|---|---|
| 10 | UT-IGSP | 1 | 98 | 26 | 31 |
| | DCDI-G | 31 | 27908 | 18731 | 9857 |
| | DCDI-DSF | 7118 | 44440 | 23798 | 5688 |
| | DCI | 41 | 1438 | 404 | 365 |
| | BACADI | 4005 | 9483 | 6284 | 1584 |
| | BACADI-L | 1076 | 1403 | 1272 | 76 |
| | CDN (MLP) | 1 | 5 | 1 | 1 |
| | CDN | 17 | 147 | 39 | 27 |
| 20 | UT-IGSP | 1 | 78 | 19 | 26 |
| | DCDI-G | 45 | 46281 | 35032 | 16068 |
| | DCDI-DSF | 23181 | 55406 | 30076 | 7200 |
| | DCI | 325 | 21414 | 4415 | 4707 |
| | BACADI | 15082 | 50737 | 26020 | 9044 |
| | BACADI-L | 2862 | 3819 | 3461 | 252 |
| | CDN (MLP) | 1 | 3 | 1 | 0 |
| | CDN | 36 | 86 | 54 | 13 |

Table 10: Uncertainty quantification on Perturb-seq datasets by sub-sampling to 80% of cells per perturbation (or 50, whichever is higher). Standard deviation reported over 5 rounds of sub-sampling. CDN is also the most robust to sub-sampling, compared to baselines. GENEPT performance is highly variable.

| | K562 gw | | | K562 es | | | RPE1 | | |
|---|---|---|---|---|---|---|---|---|---|
| Model | rank | $R_{20}$ | $r$ | rank | $R_{20}$ | $r$ | rank | $R_{20}$ | $r$ |
| LINEAR | $0.50_{\pm.00}$ | $0.17_{\pm.01}$ | $0.20_{\pm.01}$ | $0.50_{\pm.01}$ | $0.10_{\pm.01}$ | $0.28_{\pm.01}$ | $0.48_{\pm.00}$ | $0.07_{\pm.01}$ | $0.49_{\pm.01}$ |
| MLP | $0.48_{\pm.00}$ | $0.21_{\pm.00}$ | $0.20_{\pm.00}$ | $0.48_{\pm.00}$ | $0.11_{\pm.00}$ | $0.19_{\pm.00}$ | $0.53_{\pm.00}$ | $0.15_{\pm.00}$ | $0.39_{\pm.00}$ |
| GENEPT | $0.54_{\pm.09}$ | $0.28_{\pm.06}$ | $0.29_{\pm.08}$ | $0.50_{\pm.08}$ | $0.21_{\pm.09}$ | $0.39_{\pm.10}$ | $0.51_{\pm.10}$ | $0.20_{\pm.08}$ | $0.51_{\pm.07}$ |
| GEARS | $0.50_{\pm.01}$ | $0.13_{\pm.01}$ | $0.22_{\pm.01}$ | $0.51_{\pm.00}$ | $0.07_{\pm.00}$ | $0.23_{\pm.00}$ | $0.46_{\pm.00}$ | $0.07_{\pm.01}$ | $0.41_{\pm.02}$ |
| PDG | $0.49_{\pm.01}$ | $0.21_{\pm.01}$ | $0.25_{\pm.00}$ | $0.50_{\pm.01}$ | $0.16_{\pm.01}$ | $0.32_{\pm.01}$ | $0.49_{\pm.01}$ | $0.08_{\pm.01}$ | $0.47_{\pm.01}$ |
| CDN | $\mathbf{0.76}_{\pm.01}$ | $\mathbf{0.49}_{\pm.01}$ | $\mathbf{0.45}_{\pm.02}$ | $\mathbf{0.66}_{\pm.01}$ | $\mathbf{0.27}_{\pm.01}$ | $\mathbf{0.43}_{\pm.01}$ | $\mathbf{0.69}_{\pm.01}$ | $\mathbf{0.26}_{\pm.01}$ | $\mathbf{0.54}_{\pm.01}$ |

| | HepG2 | | | Jurkat | | |
|---|---|---|---|---|---|---|
| Model | rank | $R_{20}$ | $r$ | rank | $R_{20}$ | $r$ |
| LINEAR | $0.49_{\pm.00}$ | $0.06_{\pm.01}$ | $0.42_{\pm.01}$ | $0.50_{\pm.00}$ | $0.08_{\pm.00}$ | $0.27_{\pm.01}$ |
| MLP | $0.50_{\pm.00}$ | $0.09_{\pm.00}$ | $0.41_{\pm.00}$ | $0.47_{\pm.00}$ | $0.12_{\pm.00}$ | $0.31_{\pm.00}$ |
| GENEPT | $0.50_{\pm.13}$ | $\mathbf{0.19}_{\pm.08}$ | $0.45_{\pm.06}$ | $0.51_{\pm.07}$ | $0.22_{\pm.09}$ | $0.40_{\pm.11}$ |
| GEARS | $0.51_{\pm.00}$ | $0.11_{\pm.00}$ | $0.45_{\pm.00}$ | $0.52_{\pm.01}$ | $0.08_{\pm.00}$ | $0.30_{\pm.00}$ |
| PDG | $0.49_{\pm.00}$ | $0.05_{\pm.01}$ | $0.41_{\pm.01}$ | $0.49_{\pm.01}$ | $0.11_{\pm.01}$ | $0.34_{\pm.00}$ |
| CDN | $\mathbf{0.65}_{\pm.01}$ | $0.17_{\pm.01}$ | $\mathbf{0.46}_{\pm.02}$ | $\mathbf{0.65}_{\pm.01}$ | $\mathbf{0.24}_{\pm.02}$ | $\mathbf{0.43}_{\pm.01}$ |

**Perturb-seq uncertainty quantification**  Due to space limitations, we report uncertainty estimates in Table 10. Multiple baselines produce deterministic results (LINEAR, GENEPT), so instead of model randomness, we report uncertainties over the sampling of single cells. Specifically, for each perturbation with $M$ cells, we sample

$$M' = \min(M, \max(50, 0.8M)) \tag{11}$$

cells uniformly at random, repeated 5 times.

**Additional synthetic results**  Table 11 reports results on all synthetic test datasets, averaging over all interventions for each dataset.

Table 11: Intervention target prediction results on synthetic datasets, extended results. Uncertainty is standard deviation over 5 distinct datasets. Metrics are averaged over all $3N$ perturbations for a given dataset (1-3 targets).

| $N$ | $E$ | Model | Linear (Hard) | | Linear (Soft) | | Poly. (Hard) | | Poly. (Soft) | |
|---|---|---|---|---|---|---|---|---|---|---|
| | | | mAP↑ | AUC↑ | mAP↑ | AUC↑ | mAP↑ | AUC↑ | mAP↑ | AUC↑ |
| | | UT-IGSP | 0.29±.02 | 0.57±.03 | 0.30±.01 | 0.63±.02 | 0.33±.04 | 0.59±.03 | 0.32±.01 | 0.64±.02 |
| | | DCDI-G | 0.39±.04 | 0.52±.04 | 0.40±.04 | 0.51±.04 | 0.38±.03 | 0.49±.02 | 0.36±.03 | 0.49±.03 |
| | | DCDI-DSF | 0.36±.02 | 0.51±.04 | 0.40±.05 | 0.53±.05 | 0.37±.02 | 0.50±.01 | 0.38±.02 | 0.49±.03 |
| 10 | 10 | DCI | 0.55±.08 | 0.79±.04 | **0.50**±.08 | **0.70**±.05 | 0.47±.09 | 0.71±.06 | 0.41±.02 | 0.67±.01 |
| | | BACADI-E | 0.26±.03 | 0.61±.05 | 0.42±.13 | 0.63±.11 | 0.25±.03 | 0.61±.05 | 0.30±.05 | 0.61±.04 |
| | | BACADI-M | 0.23±.01 | 0.56±.03 | 0.37±.11 | 0.62±.11 | 0.22±.02 | 0.56±.04 | 0.30±.05 | 0.61±.04 |
| | | CDN (MLP) | **0.73**±.04 | **0.86**±.04 | 0.33±.03 | 0.50±.02 | 0.61±.08 | **0.77**±.07 | 0.39±.03 | 0.51±.04 |
| | | CDN (AXIAL) | **0.73**±.05 | 0.85±.03 | 0.48±.06 | 0.64±.05 | **0.62**±.09 | 0.77±.07 | **0.73**±.07 | **0.82**±.06 |
| | | UT-IGSP | 0.28±.02 | 0.56±.03 | 0.26±.01 | 0.59±.03 | 0.28±.03 | 0.58±.04 | 0.27±.01 | 0.60±.01 |
| | | DCDI-G | 0.42±.02 | 0.54±.02 | 0.40±.03 | 0.51±.04 | 0.40±.06 | 0.53±.05 | 0.39±.02 | 0.51±.03 |
| | | DCDI-DSF | 0.41±.03 | 0.52±.03 | 0.39±.05 | 0.52±.05 | 0.39±.02 | 0.51±.02 | 0.39±.03 | 0.50±.05 |
| 10 | 20 | DCI | 0.59±.03 | 0.78±.03 | **0.57**±.04 | **0.77**±.03 | 0.68±.07 | 0.82±.04 | **0.65**±.08 | **0.84**±.05 |
| | | BACADI-E | 0.34±.04 | 0.68±.03 | 0.53±.08 | 0.71±.04 | 0.33±.03 | 0.72±.04 | 0.64±.07 | 0.78±.05 |
| | | BACADI-M | 0.27±.02 | 0.63±.03 | 0.48±.09 | 0.71±.04 | 0.27±.04 | 0.64±.08 | 0.59±.09 | 0.77±.06 |
| | | CDN (MLP) | **0.78**±.06 | **0.88**±.03 | 0.38±.05 | 0.56±.05 | **0.80**±.06 | **0.89**±.03 | 0.38±.05 | 0.48±.05 |
| | | CDN (AXIAL) | 0.74±.05 | 0.86±.03 | 0.44±.08 | 0.60±.06 | 0.77±.05 | **0.88**±.03 | 0.54±.05 | 0.65±.06 |
| | | UT-IGSP | 0.15±.01 | 0.54±.01 | 0.18±.01 | 0.65±.02 | 0.20±.02 | 0.60±.02 | 0.21±.02 | 0.67±.02 |
| | | DCDI-G | 0.24±.02 | 0.49±.02 | 0.22±.02 | 0.49±.02 | 0.22±.03 | 0.50±.03 | 0.22±.02 | 0.49±.03 |
| | | DCDI-DSF | 0.24±.02 | 0.50±.02 | 0.23±.02 | 0.50±.03 | 0.22±.01 | 0.50±.02 | 0.20±.01 | 0.48±.02 |
| 20 | 20 | DCI | 0.43±.03 | 0.77±.01 | 0.45±.08 | **0.75**±.04 | 0.48±.05 | 0.76±.03 | 0.50±.02 | 0.79±.02 |
| | | BACADI-E | 0.14±.01 | 0.61±.03 | 0.25±.10 | 0.60±.06 | 0.22±.06 | 0.76±.06 | 0.30±.06 | 0.74±.05 |
| | | BACADI-M | 0.12±.00 | 0.58±.02 | 0.17±.03 | 0.60±.05 | 0.14±.02 | 0.66±.05 | 0.22±.05 | 0.71±.06 |
| | | CDN (MLP) | **0.67**±.05 | **0.87**±.02 | 0.28±.01 | 0.54±.03 | **0.60**±.06 | **0.84**±.03 | 0.34±.04 | 0.51±.04 |
| | | CDN (AXIAL) | 0.65±.03 | 0.86±.01 | **0.49**±.05 | 0.72±.03 | 0.58±.05 | 0.83±.02 | **0.65**±.02 | **0.81**±.02 |
| | | UT-IGSP | 0.14±.01 | 0.57±.01 | 0.14±.01 | 0.61±.02 | 0.20±.01 | 0.62±.02 | 0.19±.01 | 0.65±.01 |
| | | DCDI-G | 0.22±.03 | 0.49±.03 | 0.21±.02 | 0.48±.03 | 0.22±.02 | 0.49±.02 | 0.22±.02 | 0.50±.02 |
| | | DCDI-DSF | 0.21±.02 | 0.46±.03 | 0.21±.02 | 0.47±.02 | 0.23±.02 | 0.50±.03 | 0.21±.01 | 0.49±.03 |
| 20 | 40 | DCI | 0.60±.06 | 0.85±.03 | **0.50**±.02 | **0.77**±.03 | 0.56±.08 | 0.81±.05 | 0.59±.06 | **0.84**±.04 |
| | | BACADI-E | 0.23±.05 | 0.70±.04 | 0.33±.10 | 0.67±.04 | 0.29±.08 | 0.83±.04 | 0.45±.18 | 0.78±.06 |
| | | BACADI-M | 0.15±.01 | 0.64±.02 | 0.19±.04 | 0.63±.02 | 0.17±.02 | 0.70±.05 | 0.31±.10 | 0.77±.07 |
| | | CDN (MLP) | **0.78**±.04 | **0.92**±.01 | 0.29±.03 | 0.56±.03 | **0.72**±.07 | **0.88**±.04 | 0.36±.05 | 0.58±.04 |
| | | CDN (AXIAL) | 0.74±.05 | 0.90±.02 | 0.38±.03 | 0.65±.01 | 0.68±.08 | 0.87±.04 | **0.64**±.03 | 0.79±.03 |

