# OpenReview forum: "Predicting perturbation targets with causal differential networks"
_ICLR.cc/2025/Conference — Submitted to ICLR 2025_

### Official Review · Reviewer_oza9 · 2024-10-22

**Soundness:** 2
**Presentation:** 3
**Contribution:** 3
**Rating:** 5
**Confidence:** 2

**Summary:**

The paper proposes a method to predict perturbation targets in biological data using the proposed causal differential networks (CDN). The idea is to infer the causal graph from observational and interventional datasets through a pre-trained model, and then learn the mapping from features to the variables targeted by interventions. The proposed method is validated on both synthetic and real-world datasets with STOA performance. However, the reliance on accurate causal graphs and labeled data presents challenges that may need to be addressed for broader applicability.

**Strengths:**

In my view, the major contribution of this paper is to propose a potential way to scalably identify the perturbed variables, which is especially meaningful in biology tasks.

The proposed method is abundantly validated on both synthetic and real-world datasets, demonstrating the performance of CDN is non-trivial.

The presentation is generally good, despite a few expressions that might be confusing for readers without a biology background, e.g. DE.

**Weaknesses:**

Supervised learning for causal discovery is an intriguing concept. While I personally believe these two ideas are somewhat contradictory, the performance of supervised causal discovery in this paper is notable. The biggest limitation so far is the lack of discussion around the pretrained causal featurizer, which will clearly influence the identified targets. The paper mentions using a fixed pretrained model. I am not sure if there is any modification of the model across different datasets. Using the same model for every dataset is not entirely reasonable.

Another limitation is the lack of discussion of out-of-distribution. In the real-world scenario where there are no labels at all, the supervised learning-based method doesn't have any guarantee. Investigation of the performance of this scenario is appreciated and valued. (For the experiment on unseen cell lines, I am not sure I understand it correctly. Currently, I understand it as in distribution generalization rather than out-of-distribution).

**Questions:**

- What is DE in Figure~3?
- Can the authors elaborate more on "unseen cell lines"? Does that mean we have the same variables as "seen cell lines" but only different data samples, or even different distributions?
- For all datasets, exactly the same pretrained model is used or there is a modification for different datasets?

---

> ### Author Response · Authors · 2024-11-19
> **Thank you for your review**
>
> Thank you for your review and suggestions! We hope that this response provides clarity, especially with regards to the application, and we will incorporate your feedback into the writing. Please let us know if you have any other concerns.
>
> ## Generalization
>
> - On synthetic experiments, all testing datasets were unseen; and the Polynomial mechanism was entirely absent from pretraining/finetuning. We will emphasize this in the paper.
> - We constructed our Perturb-seq train/test splits to ensure that perturbations which induce similar effects are placed in the same split (Figure 3 and 310-311). Figure 5 illustrates the correlation between log-fold change induced by each perturbation, and clear clusters emerge (which we allocate to different splits).
> - Cell lines are derived from cancer tumors / other conditions (K562 myelogenous leukemia, RPE non-cancerous, HepG2 liver cancer, Jurkat acute T cell leukemia).
>
>   Biologically they are quite distinct: different genes are expressed (the supports of each distribution); among the same genes, the marginal distributions (of expression) and gene-gene relationships may differ. So it would be reasonable to consider them different data distributions. In fact, even separate experiments on the *same* cell line tend to report different results at the individual gene level [1], due to differences in lab technician, equipment, experimental protocol, etc.
>
> [1] Nadig et al. Transcriptome-wide characterization of genetic perturbations. 2024.
>
> ## Model weights
>
> We have several sets of model weights.
> 1. On synthetic data, we trained 2 models, corresponding to inverse covariance and correlation global statistics (line 239).
> 2. On Perturb-seq data, we finetuned the correlation model into 5 separate sets of weights. For unseen cell lines, we finetuned the model on the training sets of every cell line, minus the one in question (306-309). For seen cell lines, we trained an "overall" model on the union of each cell line's training sets.
> 3. For the chemical perturbations, due to limited data, we used the "overall" model without any adjustments.
>
> ## Other
>
> **"reliance on accurate causal graphs and labeled data":**
>
> Since we learn the mapping from graph representations to intervention targets, the model does not require globally accurate causal graphs; only that their *differences* are informative. This idea has been analyzed in more theoretical detail by [2] for linear Gaussian data.
>
> With regards to labeled data, pretraining on synthetic data actually contributes the most to performance (Table 7), so there is less reliance on labeled real data. In addition, the unseen cell line case + the Sci-Plex experiments demonstrate that it is not necessary to have labeled data from the intended distribution, though performance is higher if there is.
>
> [2] Belyaeva et al. DCI: learning causal differences between gene regulatory networks. Bioinformatics. 2021.
>
> **DE** refers to "differential expression," i.e. statistically significant change in the amount of a gene (details in B.1). We'll add a brief introduction and reference to the main paper. Thank you for pointing this out!

---

> > ### Comment · Reviewer_oza9 · 2024-11-26
> >
> > I thank the authors for their replies. So far I am still confused.
> >
> > > The authors mentioned in line 232 "For the causal featurizer, we used a frozen, pretrained aggregator..". The authors state that they "finetuned the correlation model", "we trained 2 models, corresponding to inverse covariance and correlation global statistics ".
> >
> > 1. What is exactly the implementation? Does the causal featurizer be finetuned or not?
> >
> > 2. What's the definition of the correlation model? I don't find out it through the paper.
> >
> > 3. What's the purpose of training two models for inverse covariance and correlation global statistics, since the correlation and inverse covariance can be obtained through the covariance matrix?
> >
> >
> > > " their differences are informative. This idea has been analyzed in more theoretical detail by [2] for linear Gaussian data."
> >
> > To me, this makes sense since for linear Gaussian data, the covariance is the sufficient statistic, which is not generally applicable.
> >
> >
> > > " pretraining on synthetic data actually contributes the most to performance (Table 7)"
> >
> > How can we conclude this conclusion from Table 7? I would appreciate it if the author could elaborate more on this statement.
> >
> >
> > Another minor question just to make sure we are on the same page.  The authors state that "Biologically they are quite distinct: different genes are expressed (the supports of each distribution); among the same genes...". Does that mean the CDN is handling OOD data?

---

> ### Author Response · Authors · 2024-11-26
> **Thank you for your questions!**
>
> Thank you for your response and questions! They are very insightful for identifying which parts of our paper are confusing.
>
> **Training procedure**: Thank you for pointing this out. Please let us know if this makes more sense, and if so, we will update the manuscript to reflect this explanation.
>
> There are two modules in Figure 2, the causal featurizer and differential network. They are trained separately, in multiple stages.
>
> 1. **Causal featurizer**: Each set of weights corresponds to a single global statistic, i.e. one for correlation, one for inverse covariance. These steps are performed by [1], and this component is not updated afterwards.
>
>     A. Causal featurizer is trained from scratch on synthetic data to predict causal graphs (inverse covariance).
>
>     B. Causal featurizer is finetuned on the same synthetic data to predict causal graphs (correlation). This is necessary because the input featurization has changed. It is helpful to finetune vs. train from scratch because i) 0s still correspond to lack of edge, and ii) the "local graph estimate" track can still be used.
>
> 2. **Differential network**: Each set of weights corresponds to a *pair* of global statistic and training data.
>
>     A. Differential network is trained from scratch on synthetic data to predict targets (inverse covariance), with frozen (inverse covariance) causal featurizer. **This model is used for synthetic experiments.**
>
>     B. Differential network is trained from scratch on synthetic data to predict targets (correlation), with frozen (correlation) causal featurizer.
>
>    C. Differential network (correlation) is finetuned on real data, 5 times for 5 data splits. **These models are used for the real experiments.**
>
> [1] Wu et al. Sample, estimate, aggregate: A recipe for causal discovery foundation models. 2024.
>
> **Table 7** ablates the contribution of model components and step 2B (training differential network on synthetic data), on performance for real data.
>
> The full procedure (on real data) is Step 1B, Step 2B, Step 2C. The ablation is Step 2C.
>
> We observe that if we do not train the causal featurizer / differential network on synthetic data, the performance of our model drops significantly (near random guessing). Our conclusion is that the real data are too small for training the model on their own (the classification task is hard to learn), and synthetic data augmentation is crucial.
>
> **Separate models for each statistic**: (Inverse) covariance maintains data scale, and [1] reports that inverse covariance performs better on synthetic data, so theoretically, these two would indeed be preferable.
>
> However, correlation is numerically more stable in high dimensions (e.g. number of genes), which goes up to $N=1000$ here. Computing covariance can lead to large values in the input (suboptimal for subsequent neural network optimization), which is why we opted for correlation on larger, biological datasets.
>
> **OOD-ness**: We believe there are two ways to interpret unseen cell lines (yes and no). With an analogy to language: suppose genes are "words," each cell is a "document," and each cell line is a category of news, e.g. "sports" vs. "popular science."
>
> 1. There are domain-specific words (genes), e.g. "football" vs. "DNA," which only appear in their respective categories (cell lines). The style of reporting might also be different, e.g. "sports" focuses on statistics of the latest games, while "popular science" focuses on telling a story.
>
>     Cell lines may express different genes and run different "gene programs." If we compute the Jaccard index (# intersection / # union) of their genes (vocabulary), the highest is 0.766 (K562 essential and Jurkat), while the lowest is 0.702 (Jurkat and HepG2). In this sense, the news categories / cell lines are out-of-distribution from each other.
>
> 2. There are also aspects that are similar, on a distributional level. For example, if the goal is to determine whether the news article is positive / negative towards its subject, we can ignore common words like "the" "and" "it", regardless of the genre. Domain-specific words for "goodness" may exist, e.g. "star player" vs. "groundbreaking discovery," but these phrases should have "sharper" distributions, e.g. high [tf-idf](https://en.wikipedia.org/wiki/Tf%E2%80%93idf).
>
>     In cells, we can ignore all genes whose relationships do not change. Across cell lines, the "effect size" of perturbations varies, e.g. how many genes are affected on average. However, the true target may have a "sharper" change, even if "by how much" differs. In this latter sense, there is information that can be shared, albeit softly.
>
> Based on these considerations, our design choice when finetuning the differential network on real data (Step 2C), was *not* to maintain the mapping of genes across datasets, i.e. the model does not know that gene X in K562 is the same as gene X in Jurkat. This was to enable generalization across cell lines.

---

### Official Review · Reviewer_br1e · 2024-11-01

**Soundness:** 2
**Presentation:** 2
**Contribution:** 1
**Rating:** 3
**Confidence:** 4

**Summary:**

This work proposes a method for predicting the (set of) variables targeted by a perturbation from observations of the variables under control and perturbation conditions. The method, called CDN, first maps both datasets to embeddings that represent the interactions of the NxN variables using a pretrained (frozen) causal discovery transformer model of prior work. Given this input representation, it then trains a shallow transformer-style prediction model in classification of the perturbation targets. The approach is evaluated on several single-cell sequencing screens involving gene and drug perturbations, as well as synthetic datasets.

**Strengths:**

The problem studied in this work is a relevant problem in the application of machine learning to biology. Contrary to several related works in the causal discovery literature, a strength of this work is its evaluation on several real-world datasets. The empirical results appear good. However, the paper does not compare to any prior specialist methods that are specifically used to predict intervention targets in biology applications, so it is hard to assess how useful the method is to the biology community.

**Weaknesses:**

-	From the perspective of the machine learning community, the method contribution is very marginal. The idea consists of extracting edge embeddings from an existing, pre-trained transformer model, and then training a 1-layer prediction layer in (multi-hot) binary classification from these features. This is a straightforward supervised learning task, and nothing about the conception or design of the method is specifically motivated by the (biological) problem at hand. The main text of the paper does not adequately motivate that the specific architecture by Ho et al. (2020) used here, apart from arguing that the function class is sufficiently descriptive (Section 3.4).
-	In particular, framing the algorithm as based on “causality” is misleading. There is no algorithmic component underlying the approach that is rooted in a mechanistic model, intervention modeling, or a causal graph. The CDN is a simple classifier that maps Xs to Ys (interaction embeddings to whether or not variables were targeted).
-	The formalization of what a “target” may not make much sense in the context of biology. In biological (dynamical) systems, changing conditional independencies may not be restricted to the variables targeted by a perturbation. This is because, if the true data-generating process is not a structural causal model, then interventions may not leave other conditional independencies invariant (see e.g. example by [A]). I also don’t agree with the statement that “it is common to assume” that the data was generated by a structural causal model (l. 50). This assumption is rarely made in computational biology and, for example, due to acyclicity assumptions of the causal graph, not commonly regarded a good assumption.
-	Motivation: The discussion around forward vs backward prediction in the introduction is confusing as a motivation of the task of perturbation target prediction. These are not the same task, and neither of them are more nor less suited for experimental design. In the “forward” task, we assume we know the targets. Why does that make these classes of “forward” methods worse at larger combinatorial search spaces than when predicting targets in a “backward” approach? In both cases, we must design new experiments in a combinatorial search space.
-	The “claims” in Section 3.4. on “Theoretical Context” are misleading/vacuous and do not provide any theoretical insight. It is not informative to say that the function class used to train the classifier (here, axial attention) is “well-specified” (i.e. complex enough) to map the covariate X to the target Y. A lot of function classes can do this (e.g. an MLP). The question is whether they will do so, for example, in the large sample limit, but no such questions are addressed or studied here.

[A] Alejandro Tejada-Lapuerta, Paul Bertin, Stefan Bauer, Hananeh Aliee, Yoshua Bengio, Fabian J Theis: Causal machine learning for single-cell genomics

**Questions:**

-	How confident can we be in the ground-truth “targets” used for the drug perturbation datasets for the evaluation?
-	“To ensure high quality labels, we filtered perturbations to those that induced over 10 differentially-expressed genes” (l. 294). This does not make the **labels** more high quality. If anything, it ensures that the considered perturbations are strong. But whether or not the target labels are accurate does not depend on the variable selection. Please provide some more information on the quality of the labels.
-	In the description of the correlation metric, what does this sentence mean?: “[…] similarity between the (ground truth) mean log- fold change of the top prediction that was observed as a perturbation, and that of the actual perturbation (given as input)” (l. 335). Specifically, what is the “top prediction that was observed as a perturbation”?
-	“LINEAR and MLP take as input the mean expression of all perturbation targets, plus the top 2000 highly-variable genes” Since the perturbation targets are what need to be predicted, what does it mean here to take as input the mean expression of all targets, which are supposedly unknown? Moreover, what does it mean to “take as input a gene”? Do you mean the mean expression of that gene again? I don’t think this baseline makes sense, because it is impossible for a method to predict the **interactions/targets** based on the mean of the variables, without any covariance/correlation information. It would make much more sense to perform classification based on either correlation/covariance information of all NxN variables or the pre-trained edge embeddings used by CDN.

---

> ### Author Response · Authors · 2024-11-19
> **Thank you for your review**
>
> Thank you for your review and questions. We hope this response clarifies your confusion, and we look forward to discussing with you!
>
> **"the paper does not compare to any prior specialist methods that are specifically used to predict intervention targets in biology applications"** As we mention in Section 4.1 (338-361), the majority of baselines on our biological experiments *are* specialist methods and the current state-of-the-art in perturbation modeling:
> - PDGrapher is designed for this exact task, i.e. predicting perturbation targets.
> - GEARS is an established baseline for unseen perturbation effect prediction, whose goal is to "guide the design of perturbational experiments"
> - GenePT is a recent gene/cell "foundation model" that reports state-of-the-art results on a variety of related tasks.
>
> **algorithmic novelty**
>
> We will improve the writing to highlight the novelty of our architecture.
>
> Ho et al. (2020) is commonly cited for the idea of axial attention, e.g. in MSA Transformer for protein sequence modeling [1], but the architecture we use is quite different.
> - The original "Axial Transformer" was designed for images, with considerations for image patch size, spatial locality, color channels, etc.
> - In our case, the "axial attention based model" operates over paired graph adjacency matrices, so it is invariant to node labeling order, but does consider the ordering of observational vs. interventional graphs. Axial attention was a natural choice for scaling to large adjacency matrices with $O(N^2)$ entries.
> - Pooling over incoming edges was also inspired by the intuition that exogenous interventions sever or alter the relationships between parents and target nodes.
>
> [1] Rao et al. MSA Transformer. ICML 2021.
>
> **nothing about the conception or design of the method is specifically motivated by the (biological) problem at hand**
>
> From a causality perspective, the task of unknown intervention prediction has been "solved" in many theoretical settings. However, existing algorithms 1) impose strict assumptions regarding the data generation process, 2) are analyzed in the infinite data limit, and 3) scale poorly to >100 variables (lines 141-152).
>
> While not limited to biological applications, certain challenges posed by transcriptomics include: 1) it is unclear what assumptions should hold, if they can even be enumerated; 2) each perturbation has limited data; and 3) measurements are over hundreds or thousands of variables (lines 84-86).
>
> Amortized causal discovery algorithms are designed to address (1) by training over large numbers of datasets, which represent diverse data generating processes (Section 2.3). To address data sparsity and scalability, [2] only requires ~200 data points for high causal discovery performance, and runs on up to 1000 variables in seconds.
>
> [2] Wu et al. Sample, estimate, aggregate: A recipe for causal discovery foundation models. 2024.
>
> **"no algorithmic component underlying the approach is rooted in a mechanistic model, intervention modeling, or a causal graph"**
>
> The causal featurizer is an amortized causal discovery algorithm that infers graph representations from each cell population. While biology largely lacks "ground truth" causal graphs for evaluation, the predicted graphs from [2] have been validated on simulated mRNA data (directed graphs) and on physical protein-protein interaction graphs (undirected graphs), which inspire its application here.
>
> With regards to interventions, many classical causal discovery algorithms are dedicated towards the observational setting, in which certain graph topologies are still identifiable, under many conditions, see [3]. Here, the outputs of the FCI algorithm are input to our model, which was trained to reconcile errors and inconsistencies, as real data are noisy and often misspecified. FCI estimates are asymmetric and represent rich information, e.g. ancestry or confoundedness.
>
> [3] Sprites, Glymour, and Scheines. Causation, prediction, and search. 2001.
>
> **"The CDN is a simple classifier that maps Xs to Ys (interaction embeddings to whether or not variables were targeted)"**
>
> On a high level, mapping datasets X to graphs G and/or intervention targets Y *is* the goal of causal discovery. The key lies in how this mapping is learned. Our insight is that modeling putative causal graphs provides useful inductive biases towards this predictive task.
>
> In Table 7, we ablated removing the pretrained amortized causal discovery weights and found that the resultant model (which is a "simple classifier that maps Xs to Ys") performs near random.
>
> | | K562 gw | K562 es | RPE1 | HepG2 | Jurkat |
> |-|-|-|-|-|-|
> | CDN | 0.72 | 0.56 | 0.59 | 0.55 | 0.62 |
> | classifier | 0.50 | 0.51 | 0.49 | 0.52 | 0.54 |
>
> Therefore, we concluded that causal structure prediction *is* a useful pretraining objective for this task.

---

> ### Author Response · Authors · 2024-11-19
>
> **what a “target” may not make much sense in the context of biology**
>
> We agree that in biology, the concept of a "target" can be hard to define. In this work, we loosely consider targets as molecules which are *physically impacted* by perturbation, rather than any requirement on conditional independencies.
> - Since CRISPRi targets promoters of expressed genes, we treat the intended gene as the "target" (line 291-292).
> - In the case of drugs, the "target" is less clear in the context of transcriptomics, since drugs bind physically to protein targets instead of genes. However, prior work [4] has shown that a drug's mechanism of action can be inferred from transcriptomic readouts, presumably due to feedback mechanisms, which is why we consider the protein target's gene counterpart as the "target" (293).
>
> [4] Iorio et al. Discovery of drug mode of action and drug repositioning from transcriptional responses. PNAS 2010.
>
> **"I also don’t agree with the statement that 'it is common to assume' that the data was generated by a structural causal model"**
>
> We agree that in the mechanistic modeling literature, dynamical systems are more common + correct representations of the relationships between biological variables.
>
> However, we intended to convey that the SCM formalism is a *simplifying* assumption, often made to reduce the space of plausible explanations. For example, [5] is an early work that demonstrated that causal methodologies can automatically recapitulate known signaling pathways, and [6] is a more recent work that used causal models to narrow down the space of drug repurposing candidates. Thus, following the classic statistics aphorism, we acknowledge that this formulation may be *wrong*, but as our experiments show, we hope that the resultant model may still be *useful*.
>
> [5] Sachs et al. Causal Protein-Signaling Networks Derived from Multiparameter Single-Cell Data. Science. 2005.
> [6] Belyaeva et al. Causal network models of SARS-CoV-2 expression and aging to identify candidates for drug repurposing. Nat Commun. 2021.
>
> **perturbation targets**
>
> - **Label quality:** Due to standard quality control procedures, the identity of each cell's perturbation target is one of the most reliable readouts from Perturb-seq experiments, as it is non-quantitative. When we refer to "noise" in transcriptomic data, we typically refer to the variability in the amount of each gene, rather than sequencing error (modern sequencing machines have error rates of 0.1-0.5% [7], and library design takes this into account).
>
>   More specifically [8]: In Perturb-seq (Figure 3), single cells are annotated with a unique barcode after they are isolated into individual droplets. Thus, each "read," i.e. individual instance of a gene / intended perturbation, is associated with one particular cell. Cells that cannot be confidently mapped to a single perturbation (doublets, unviable cells, empty droplets) are discarded prior to any analysis.
>
> - **Strong perturbations:** Since the goal of our framework is to achieve a particular phenotypic profile, we focus on strong perturbations to ensure that the desired phenotype is indeed *different* from the control. In this regard, it appears that "label quality" is a poor word choice, and we will update the paper to reflect this – thank you for pointing this out.
>
> [7] Stoler and Nekrutenko. Sequencing error profiles of Illumina sequencing instruments. NAR Genomics and Bioinformatics, Volume 3, Issue 1, March 2021, lqab019.
>
> [8] Luecken and Theis. Current best practices in single‐cell RNA‐seq analysis: a tutorial. Mol Syst Biol (2019) 15: e8746.
>
> **"forward" vs. "backward" modeling**
>
> To the best of our knowledge, there does not currently exist a genome-scale experiment to combinatorially screen genetic perturbations, nor to parallelize across different cell lines (they may have different growth requirements, characteristics, etc.).
>
> Given the experiments that we do have, we introduced the concepts of "forward" and "backward" modeling as two different strategies to extract information from perturbation screens, to inform iterative experimental design or biological understanding. In both cases, the available **training** data are **identical.** In the "forward" case, we train a model to map `X_obs, Y` to `X_int`. In the "backward" case, we map `X_obs, X_int` to `Y`.
>
> Given these trained models, there are two applications that could be of interest.

---

> > ### Author Response · Authors · 2024-11-19
> >
> > 1. If we wish to simulate unseen perturbations to a "virtual cell," then the "forward" approach is clearly superior.
> > 2. On the other hand, we may have sets of diseased and healthy tissue, and we would like to know what genes are to blame. Here, we *could* run the "forward" model "O(N)" times, where N is the number of candidate sets. Alternatively, we could run the "reverse" model once.
> >
> > We agree that the two tasks are different. The "forward" model was designed with the former task in mind. However, we include these baselines here (e.g. GEARS, GenePT) because there has been substantially more research dedicated towards this task, and these models are more well established in the context of experimental design.
> >
> > **correlation metric:** To reiterate, not all baselines can output predictions for each candidate gene, due to knowledge graph coverage. GEARS has the highest rate of missing candidates (Figure 8, page 19), as the majority of perturbations are missing at least 10% of their candidate genes in the GEARS graph. To account for this discrepancy, we measure the "closeness" of each model's top prediction to our target cell state.
> >
> > As an example: Suppose the candidates are $\{ x_1, x_2, x_3 \}$, where $x_2$ is the ground truth, and $y_i$ indicates the mean log-fold change in expression after perturbing each candidate $x_i$.
> > - Model 1 predicts $P(x_1) > P(x_2)$
> > - Model 2 predicts $P(x_3) > P(x_2) > P(x_1)$
> > - Model 3 predicts $P(x_2) > P(x_1)$
> >
> > We report $r(y_1, y_2)$ for Model 1; $r(y_3, y_2)$ for Model 2; and $r(y_2, y_2)$ for Model 3.
> >
> > **"misleading/vacuous" claims:**
> >
> > We agree that many architectures can approximate arbitrary functions, which is why we wanted to illustrate that our axial attention also has sufficient capacity to perform this task. The AdamW optimizer provably converges [9] (and is widely used in modern LLMs). In terms of generalization, the model converges to ~0.9+ validation AUC (held-out graphs), so even though we do not provide formal guarantees for generalization in the large data limit, this task does seem learnable.
> >
> > [9] Loshchilov and Hutter. Decoupled weight decay regularization. ICLR 2019.
> >
> > Recent work [10] has also argued that with respect to generalization / identifiability, amortized causal discovery algorithms trade explicit assumptions for implicit ones, encoded by the training data.
> >
> > - Our real finetuning data (training sets of Perturb-seq data) reflect the assumptions that may be found in similar (biological) data. We constructed our Perturb-seq train/test splits to ensure that perturbations which induce similar effects are placed in the same split (Figure 3 and 310-311). Figure 5 illustrates the correlation between log-fold change induced by each perturbation, and clear clusters emerge (which we allocate to different splits).
> > - Our synthetic pretraining data contain a variety of linear and non-linear data, as well as both Erdos-Renyi and scale-free graphs. All testing datasets were unseen; and the Polynomial mechanism was entirely absent from pretraining/finetuning. We will emphasize this in the paper.
> >
> > Therefore, while we do not provide formal guarantees in the classical sense, we expect that the diverse synthetic data + real finetuning data would convey implicit assumptions relevant to the end task.
> >
> > [10] Montagna et al. Demystifying amortized causal discovery with transformers. 2024.
> >
> > **additional questions regarding linear/MLP baselines**
> >
> > We will expand the descriptions in the manuscript. Thank you for pointing this out.
> >
> > These two baselines are designed to quantify the "predictability" of the target from expression alone, so we don't particularly expect them to be competitive. With respect to the "correlation" baseline, see above (we find that causal structure learning is an essential pretraining objective).
> >
> > Given a fixed cell line, we identify the top 2000 HVGs across all cells. We take the union of these genes, and the ground truth perturbation targets (recall, this baseline is only to quantify predictability, not act as a prospective model). This yields a set of ~2-3k genes $S$. For each perturbation $i$, we subset the cells to those perturbed by $i$, and subset the genes to $S$ to obtain $X_i$. Let $y_i$ denote the one-hot vector with a 1 at $i$ and 0 elsewhere. The goal of these models is to predict $y_i$ from $X_i$ (another instance of a "simple classifier").

---

> ### Comment · Reviewer_br1e · 2024-11-22
> **Thanks**
>
> Thank you for your replies, I remain with my current rating, since the answers do not address my concerns. Your description of the algorithmic novelty of the approach does not provide any additional evidence to me that there is more contribution. I also still remain with my view that nothing about this approach is causal, or mechanistic, so I find it misleading to use the term for a supervised learning method.
>
> Below, I reply to some other comments:
>
> > GEARS is an established baseline for unseen perturbation effect prediction,
>
> To my knowledge, this is not true. GEARS is primarily designed to predict perturbation effects, not perturbation targets.
>
>
> >  Due to standard quality control procedures, the identity of each cell's perturbation target is one of the most reliable readouts from Perturb-seq experiments, as it is non-quantitative.
>
> My question concerned the drug perturbation data, not the perturb-seq (gene perturbation) data. I am unsure about how confident we can be in the targets there.
>
> > We agree that many architectures can approximate arbitrary functions, which is why we wanted to illustrate that our axial attention also has sufficient capacity to perform this task. The AdamW optimizer provably converges [9]
>
> This is confusing in two ways. First, illustrating that axial attention has sufficient capacity is not an informative claim since, as stated above, “many architectures can approximate arbitrary functions”. Second, it is irrelevant whether AdamW converges, because the question is whether it converges to the right solution (this is a non-convex optimization problem)

---

> > ### Author Response · Authors · 2024-11-22
> > **Thank you for your response!**
> >
> > Thank you for your reply and for sharing your perspective!
> >
> > **Regarding supervised / amortized causal discovery**: while this line of work is relatively newer, there has been an increasing number of works in this area, and we are not the first to propose a supervised approach. Specifically, other supervised causal discovery algorithms (which map $X$ to $G$) include the following:
> >
> > 1. Li et al. Supervised Whole DAG Causal Discovery. 2020.
> > 2. Lorch et al. Amortized Inference for Causal Structure Learning. NeurIPS 2022.
> > 3. Ke et al. Learning to Induce Causal Structure. ICLR 2023.
> > 4. Mahajan et al. Zero-Shot Learning of Causal Models. 2024.
> >
> > Montagna et al. [5] studies [3] in detail, to find that these models still obey classical identifiability theory and formulates assumptions implicitly through the training set.
> >
> > 5. Montagna et al. Demystifying amortized causal discovery with transformers. 2024.
> >
> > **Regarding GEARS**:
> > - "GEARS is an established baseline for unseen perturbation effect prediction,"
> > - "To my knowledge, this is not true. GEARS is primarily designed to predict perturbation effects, not perturbation targets."
> >
> > We believe we are in agreement, in that GEARS predicts perturbation *effects.* To the best of our knowledge, PDGrapher is currently the closest "specialist method" for perturbation *target* prediction from single cell mRNA data.
> >
> > We benchmarked GEARS because from the application perspective, it can also be *used* as an oracle to prioritize targets, e.g. as Huang et al. [6] proposes, or as the authors of GEARS write in the Discussion:
> >
> > > GEARS can guide the design of new screens by **identifying perturbations** that maximize information gained and minimize experimental costs
> >
> > 6. Huang et al. Sequential Optimal Experimental Design of Perturbation Screens Guided by Multi-modal Priors. 2023.
> >
> > **Drug perturbation targets**
> >
> > The drug targets we consider were collated by the authors, and are each are well-documented in the literature.
> >
> > - [Volaserib](https://go.drugbank.com/drugs/DB12062) has been [co-crystallized with its target](https://www.rcsb.org/structure/3FC2) and a number of works confirm [specificity for PLK1](https://pmc.ncbi.nlm.nih.gov/articles/PMC8166193/).
> > - Palbociclib has also been [co-crystallized with its target](https://www.rcsb.org/structure/5L2I) and its mechanism is also [well-documented by the FDA](https://www.accessdata.fda.gov/drugsatfda_docs/label/2019/207103s008lbl.pdf).
> > - Infigratinib: [DrugBank](https://go.drugbank.com/drugs/DB11886) reports that Infigratinib binds FGFR with nano-molar affinities, citing [FDA documentation](https://www.accessdata.fda.gov/drugsatfda_docs/label/2021/214622s000lbl.pdf)
> > - Doxorubucin: [DrugBank](https://go.drugbank.com/drugs/DB00997) has 7 citations for TOP2A + [FDA](http://www.accessdata.fda.gov/drugsatfda_docs/label/2015/050718s048lbl.pdf) writes that "DOXIL is an anthracycline topoisomerase II inhibitor"
> > - Nintedanib is similarly well-documented in [DrugBank](https://go.drugbank.com/drugs/DB09079)
> >
> > **Re: computability** – In general, it is true that there are many universal approximators. However, it is not true that with finite precision, fixed width/depth, any function/task is learnable. For example, [7] proves that "A single layer of standard multi-headed self-attention cannot compute Match3 [a toy function of a triplet] unless its number of heads H or embedding dimension m grows polynomially in N."
> >
> > 7. Sanford et al. Representational Strengths and Limitations of Transformers. NeurIPS 2023.

---

### Official Review · Reviewer_kqT8 · 2024-11-02

**Soundness:** 2
**Presentation:** 2
**Contribution:** 1
**Rating:** 3
**Confidence:** 4

**Summary:**

The authors tackle the problem of identifying the intervention target from two (namely, observational and interventional) datasets on e.g., single cells. Supervised causal learning approaches are used to estimate two structures separately for each dataset, and then target is identified by comparing the two estimated structures.

**Strengths:**

- The use of supervised causal discovery in biology field is interesting -- though without solid asymptotic guarantee, supervised method usually impose priors about the domain to the model (which is exactly the case for biology data; while the priors can usually be hard to formalize) to improve the empirical performance.

- The experiments are relatively comprehensive.

**Weaknesses:**

1. The motivation is quite unclear, making the contribution also questionable:
   - To me the problem of identifying intervention target (at least in the presence of latent variables) is too easy and standard: theoretically one variable is intervened iff given any other set of variables, its conditional distribution changes between the two distributions, i.e., there is an direct edge between the domain index variable and it.
   - The authors claim that existing methods "scale poorly", "must search combinatorial spaces": I don't quite get that. According to above, one only need to locally learn the Markov boundary of the domain index variable (which is a binary variable 0/1 indicating for each sample which dataset it is from in the pooled dataset). Since we usually assume that the domain index is a root variable (interventions are applied exogenously), such MB consists of the children and spouses, and then some trivial CI tests within the MB can reveal the true children. The MB can be learned very efficiently, using e.g., graph LASSO or its nonlinear variants.
   - Speaking of efficiency, instead, what the authors proposed (to learn two graphs separately, and then compare the two only for learning the intervention target) seems too overkill.
   - So could the authors give more explanations for where and how the proposed method is better than existing ones?

2. For a identifiability guarantee, a formal list of assumptions (e.g., which kind of faithfulness is needed) are expected.

3. More details and insights about the usage of supervised model is needed:
   - The choice of supervised learning isn't merely due to its availability. We choose it because it can leverage abundant data from either real world or synthetics, and hopefully gain some prior knowledge (which can be hard to formalized) about the domain to the model. So, more elaboration on these are expected: how authors simulate the data or collect real-world data for this specific case, for the model's reliability and generalizability?
   - Also, how are the two graphs learn separately? Are there any constraints to ensure they don't vary too much? How to solve the conflicts between the two learned graph, e.g., some difference that cannot be explained by any intervention?
   - Please explain why it is necessary to learn two graphs separately. It appears way too overkill.

4. More literature review are expected (especially on the early developments):
   - For (generally) identifying changes:
     - Kevin D Hoover. The logic of causal inference: Econometrics and the conditional analysis of causation.
Economics & Philosophy, 6(2):207–234, 1990.
     - Jin Tian and Judea Pearl. Causal discovery from changes. arXiv preprint arXiv:1301.2312, 2001.
     - Whitney K Newey and James L Powell. Instrumental variable estimation of nonparametric models. Econometrica, 71(5):1565–1578, 2003.
     - Kevin B Korb, Lucas R Hope, Ann E Nicholson, and Karl Axnick. Varieties of causal intervention. In PRICAI 2004: Trends in Artificial Intelligence: 8th Pacific Rim International Conference on Artificial Intelligence, Auckland, New Zealand, August 9-13, 2004. Proceedings 8, pp. 322–331. Springer, 2004.
      - Biwei Huang, Kun Zhang, Jiji Zhang, Joseph Ramsey, Ruben Sanchez-Romero, Clark Glymour, and Bernhard Schölkopf. Causal discovery from heterogeneous/nonstationary data. Journal of Machine Learning Research, 21(89):1–53, 2020.
      - Amin Jaber, Murat Kocaoglu, Karthikeyan Shanmugam, and Elias Bareinboim. Causal discovery from soft inter- ventions with unknown targets: Characterization and learning. Advances in Neural Information Processing Systems, 33:9551–9561, 2020.
   - For supervised causal discovery:
      - Isabelle Guyon. Cause-effect pairs kaggle competition, 2013. URL https://www. kaggle. com/c/cause- effect-pairs, pp. 165, 2013.
      - Li, Hebi, Qi Xiao, and Jin Tian. "Supervised whole dag causal discovery." arXiv preprint arXiv:2006.04697, 2020.
      - Dai, Haoyue, et al. "Ml4c: Seeing causality through latent vicinity." Proceedings of the 2023 SIAM International Conference on Data Mining (SDM). Society for Industrial and Applied Mathematics, 2023.

**Questions:**

See in "weaknesses".

---

> ### Author Response · Authors · 2024-11-19
> **Thank you for your review**
>
> Thank you for your extensive comments and recommendations! We hope that this and the general response provide clarity on your concerns, and we look forward to discussing with you!
>
> ## Markov boundary + CI
>
> We implemented a proof of concept of this approach to better illustrate its advantages/drawbacks. We first perform graph lasso with scikit-learn using default parameters. To ensure comparability to existing metrics, instead of discrete CI decisions, we compute the mutual information of MB variables with the domain variable.
>
> **Advantages**: Overall, the performance is reasonable, and the method is generally fast on small data.
>
> On the synthetic data, we report mAP / AUC with ours and the next best baseline (from Table 4). MB+CI performs similarly to CDN on Polynomial, while CDN is much better on Linear. This baseline is indeed fast on the synthetic data (averaging ~20s / dataset, vs. CDN's 1s for MLP, 39s for Axial).
>
> | | Linear (1) | Poly. (1) | Linear (3) | Poly. (3) |
> |-|-|-|-|-|
> | MB+CI | 0.47 / 0.64 | **0.66** / **0.80** | 0.58 / 0.63 | 0.73 / 0.78 |
> | DCI | 0.42 / 0.75 | 0.39 / 0.72 | 0.63 / 0.80 | 0.53 / 0.69 |
> | CDN | **0.73** / **0.88** | 0.60 / **0.80** | **0.87** / **0.91** | **0.74** / **0.80** |
>
> On the real Sci-Plex data, this simple baseline performs well on 2/6 cases, and underperforms in the rest. A larger evaluation set might reveal more about the failure modes of both.
>
> | | A-Infig | A-Nint | A-Palb | T-Doxo | T-Palb | T-Vola |
> | --- | --- | --- | --- | --- | --- | --- |
> | MB+CI | 0.72 | 0.11 | 0.22 | **0.70** | 0.01 | **0.97** |
> | PdG | 0.71 | 0.34 | 0.39 | 0.05 | 0.34 | 0.42 |
> | Ours | **0.88** | **0.43** | **0.58** | 0.05 | **0.54** | 0.64 |
>
> **Disadvantages:**
>
> - For theoretical investigation, it makes complete sense to pre-specify the scope of each algorithm or model. In terms of practical application, however, these assumptions are not known in advance.
> - Practically speaking, mismatched assumptions lead to difficult / unstable numerical optimization. Graphical lasso converges within seconds on linear datasets, but when given a budget of 1000 iterations, graphical lasso fails to converge on ~15% of our synthetic test sets, and after 10,000 iterations, only ~2% more converge. Numerical stability (`NaN` values) also became an issue in ~20 synthetic cases.
> - This method was very slow on Perturb-seq data, sometimes requiring the same amount of time per perturbation (minutes), as CDN takes on the whole dataset (cell line). For this reason, we chose not to benchmark MB+CI on the full Perturb-seq data during the rebuttal period.
> - The two stage procedure can cause error propagation, and the outputs of the first stage are also subject to hyperparameters like the sparsity regularization weight.
>
> ## Identifiability and training data
>
> Recent work [1] has argued that amortized causal discovery algorithms trade explicit assumptions for implicit ones, encoded by the training data (while still abiding by classical identifiability theory, e.g. linear Gaussian and invertible data are non-identifiable).
>
> - Our real finetuning data (training sets of Perturb-seq data) reflect the assumptions that may be found in similar (biological) data. We constructed our Perturb-seq train/test splits to ensure that perturbations which induce similar effects are placed in the same split (Figure 3 and 310-311). Figure 5 illustrates the correlation between log-fold change induced by each perturbation, and clear clusters emerge (which we allocate to different splits).
> - Our synthetic pretraining data contain a variety of linear and non-linear data, as well as both Erdos-Renyi and scale-free graphs. All testing datasets were unseen; and the Polynomial mechanism was entirely absent from pretraining/finetuning. We will emphasize this in the paper.
>
> Therefore, while we do not provide formal guarantees in the classical sense, we expect that the diverse synthetic data + real finetuning data would convey implicit assumptions relevant to the end task.
>
> [1] Montagna et al. Demystifying amortized causal discovery with transformers. 2024.

---

> > ### Author Response · Authors · 2024-11-19
> >
> > ## Learning two graphs
> >
> > It would seem intuitive that jointly predicting a single graph is more efficient than predicting two separate graphs. At the same time, there is no requirement for these two graphs to be globally correct, only that their differences reflect the intervention targets. This idea was first proposed by DCI [2], which analyzes this concept in more theoretical detail (for linear Gaussian).
> >
> > Additional motivations for this design choice include:
> > - Empirically, it was difficult to balance the graph learning and intervention prediction objectives. Methods that learned a joint graph (DCDI, BaCaDI) performed much worse at predicting targets on synthetic data than DCI or the proposed MB+CI (Table 4, 483-485).
> > - Amortized causal discovery algorithms are very fast at inference, so the time cost of running the model twice is minimal. In terms of graph quality, previous studies have shown that they meet or exceed methods that can jointly learn graphs [3, 4].
> > - While we do not explicitly encourage the predicted graphs to be similar, given the sparsity of interventions, the input featurizations (e.g. correlation matrices, local estimates) tend to be quite similar, so the graph outputs also do not vary significantly.
> > - The model is learned end to end, with the same amortized model run twice. During training, the model converges to ~0.9+ validation AUC (held-out graphs), so these differences do seem to be learnable.
> >
> > [2] Belyaeva et al. DCI: learning causal differences between gene regulatory networks. Bioinformatics. 2021.
> >
> > [3] Lorch et al. Amortized inference for causal structure learning. NeurIPS 2022.
> >
> > [4] Wu et al. Sample, estimate, aggregate: A recipe for causal discovery foundation models. 2024.
> >
> > ## Literature review
> >
> > We are happy to expand upon our literature review. We have already cited Jaber et al. 2020, as it is one of the earliest papers on this specific topic, but will include more discussion about causal discovery in general. Thank you for the recommendations!

---

> ### Comment · Reviewer_kqT8 · 2024-11-25
>
> Thank the authors for the response. My concerns are still not addressed: just to predict the intervention target, 1) to learn the Markov blanket is theoretically sufficient, and 2) to use supervised method to learn two graphs is too overkill.
>
> The authors experiments above further support my concern. MB learning can always be tuned more accurate (there are ones for nonlinear relations instead of just graph LASSO or graphical LASSO). In terms of efficiency, one don't need to learn the whole MB; there are variants to learn only the MB of a local variable. In this sense, I have no excuse to believe that separately learning two graphs is more efficient than a local MB learning.

---

> > ### Author Response · Authors · 2024-11-25
> > **Thank you for your response**
> >
> > Thank you for your response!
> >
> > The main motivation of our amortized inference method is that, while many frameworks are theoretically sufficient, it is unclear how they should be tuned on real data. For example, in the "unseen cell line," we do not have labeled data from the target distribution prior to running inference, so there is no validation set for model selection. In principle, one could obtain sets of hyperparameters from other cell lines, but it is unclear how to combine them, or which to prioritize. Amortized inference provides a means to "softly" incorporate assumptions from a variety of potentially related data.
> >
> > Finally, to reiterate, our method does not require learning two globally accurate graphs either (similar to DCI); it only requires that the differences are informative. We also found that the baselines which focused on global graph recovery performed worse, empirically, at identifying targets.

---

### Official Review · Reviewer_TN1g · 2024-11-03

**Soundness:** 3
**Presentation:** 3
**Contribution:** 2
**Rating:** 6
**Confidence:** 3

**Summary:**

The authors focus on the problem of Identifying the direct targets of interventions in single cell perturbation data (perturb-seq). The challenge consists in filtering solely the direct targets out of a large number of downstream affected genes. The proposed approach, namely causal differential networks (CDN), first extract the causal networks underlying unperturbed and perturbed data. The networks are then compared with an axial-attention based classifier, so that to identify nodes (genes) that are the direct targets of the intervention. Finally, the authors assess CDN's performances on five perturb-seq datasets, as well as on data from two chemical perturbation studies.

**Strengths:**

This work addresses an interesting problem, i.e., the identification of direct targets in perturbation experiments.  The manuscript is well structured and written, and an extensive evaluation is presented both on real-world and simulated data.

**Weaknesses:**

The novelty of the CDN method is debatable, being the direct combination of two other already published algorithms, i.e., an amortized causal discovery model (Wu et al.,2024) and the axial attention-based classifier (Ho et al., 2020).

Interventions may directly affect genes other than their intended targets (off target effects). However, it is not clear if this issue was taken into consideration during the analysis of real-world data.

Experiments would benefit from a comparison against a trivial classifier, so that to establish a truly minimal baseline.

**Questions:**

I would like to ask the authors to address the weaknesses highlighted above:
- please elaborate further on the novelty of your proposed approach. Did you devise any algorithmic improvement? Or is CDN solely the combination of previously published algorithms?
- Can you explain whether the current analysis setting would be robust against biases introduced by off-target effects?
- As reported in Table 6, the direct target of many interventions can be trivially identified as the gene with the largest fold-change in expression. It would be useful to include a trivial classifier in all computational comparisons that simply considers the first n genes with the largest log fold change as direct targets (n is a user-set parameter). Such a classifier would help gauging what additional contribution complex causal approaches provide with respect to a very simplistic solution.

---

> ### Author Response · Authors · 2024-11-19
> **Thank you for your review**
>
> Thank you for your review and comments! We hope that this response clarifies some of your confusion.
>
> **Algorithmic contribution:** We will improve the writing to highlight the novelty of our architecture. Ho et al. (2020) is commonly cited for the idea of axial attention, e.g. in MSA Transformer for protein sequence modeling [1], but the architecture we use is quite different.
> - The original "Axial Transformer" was designed for images, with considerations for image patch size, spatial locality, color channels, etc.
> - In our case, the "axial attention based model" operates over paired graph adjacency matrices, so it is invariant to node labeling order, but does consider the ordering of observational vs. interventional graphs. Axial attention was a natural choice for scaling to large adjacency matrices with $O(N^2)$ entries.
> - Pooling over incoming edges was also inspired by the intuition that exogenous interventions sever or alter the relationships between parents and target nodes.
>
> **Off-target effects:** In terms of modeling, we output a separate binary score per gene, so in principle, the model is able to predict multiple targets per condition. In the synthetic data, the number of targets varies from 1-3, and we don't impose any additional inference-time constraints.
>
> In terms of the biological data:
> - Many of the Sci-Plex drugs are annotated with off-targets. However, these drugs did not result in differential expression of any annotated on/off target, so we omitted them from the evaluation. These drugs with off-target effects also tended to have smaller effect sizes, median 59 differentially expressed genes vs. 373.
> - For the Perturb-seq datasets, CRISPRi tends to be quite specific. Replogle et al 2022. reports 92.5% on target rate, and "neighbor gene knockdown was **not** enriched in perturbations with a negligible growth defect that produced a transcriptional phenotype," i.e. in our subset.
>
> **Table 6 and trivial baseline:** On the Sci-Plex data, we can compare to classic differential expression analysis, where we rank targets by their adjusted p-value (Wilcoxon ranked sum test, BH correction). Interestingly, the targets that DE / our method predict better seem to be mutually exclusive, so a larger evaluation set might reveal more about the failure modes of both. This also suggests that these two approaches could be complementary, as one focuses primarily on pointwise differences, while the other focuses more on pairwise changes.
>
> | | A-Infig | A-Nint | A-Palb | T-Doxo | T-Palb | T-Vola |
> | --- | --- | --- | --- | --- | --- | --- |
> | DE | 0.12 | **0.63** | 0.28 | **0.93** | 0.23 | **0.88** |
> | PdG | 0.71 | 0.34 | 0.39 | 0.05 | 0.34 | 0.42 |
> | Ours | **0.88** | 0.43 | **0.58** | 0.05 | **0.54** | 0.64 |
>
> Re: large numbers of "trivial" perturbations – Our subset of Perturb-seq data are somewhat "artificial" in that the intended target is often that with the largest log-fold change, due to selection bias and CRISPRi's specificity (above). Since the goal of our framework is to achieve a particular phenotypic profile, we subsetted to strong perturbations, to ensure that the desired phenotype is indeed *different* from the control. For example, the raw K562 genome-wide data actually had 9,851 perturbations, but the vast majority had little effect. After filtering, we were only left with 1,767 perturbations (678 test), which by design, tend to report the intended target as the gene with the largest change.

---

> > ### Comment · Reviewer_TN1g · 2024-11-27
> >
> > I would like to thank the authors for their efforts in answering my concerns. Overall, I still find the methodological contribution of the paper to be not particular outstanding. As I remarked in my original comments, CDN is at its core the combination of two previously published methods, even if the Axial Transformer did undergo modifications for better modeling biological data. On the other hand, the comparison reported in Table 6 raise even more concerns: as the authors note, "the targets that DE / our method predict better seem to be mutually exclusive". Did the author investigate the reasons behind this behaviour? This is particularly import for biologists / bioinformaticians interested in applying CDN on novel data: which predictions should be taken into account (DE or CDN), when the ground truth  is not known?
> > In summary, I do not have elements at this stage for modifying my evaluation on the work.

---

> ### Author Response · Authors · 2024-11-27
> **Thank you for your response!**
>
> Thank you for your time and insights!
>
> Regarding the Sci-Plex results, we suspect that DE might have performed particularly well on doxorubicin, since its target TOP2A is directly involved in transcription (the only case out of the six). As a result, we might expect to see high transcriptomic response as a result of its inhibition.
>
> On a limited screening budget, DE does miss the target entirely in half the cases, so perhaps it would be preferable to test half of the top predictions from each approach, to provide better coverage.

---

### Author Response · Authors · 2024-11-19
**Note to all reviewers**

Dear reviewers,

Thank you for reading our paper and providing thorough feedback. We would like to highlight three aspects that we believe are central to the motivation behind this work.

1. Biological understanding has long been a motivation for causal modeling [1]. In this work, we aimed to bridge the gap between classical ideas and pragmatic application. Specifically, our primary focus was to facilitate experimental design and biological interpretation, by predicting variables responsible for changes to cellular systems. This concept is naturally realized within a causal discovery framework, which while imperfect, provides useful inductive biases for approaching the problem.

[1] Sachs et al. Causal Protein-Signaling Networks Derived from Multiparameter Single-Cell Data. Science. 2005.

2. Identifying unknown intervention targets has been solved in many theoretical settings, but it remains a challenge and is underexplored in real applications. As we note in Section 2.2, this task has been studied extensively in the causality literature. However, guarantees (e.g. of identifiability) often require infinite data, with parametric assumptions regarding the noise and/or causal mechanisms. In single-cell perturbation experiments, data are limited, while graphs are large, and assumptions unknown. To address these issues, we draw inspiration from amortized / supervised causal discovery. These pretrained models, coupled with our attention-based architecture, achieves state-of-the-art performance across a number of real datasets.

3. We envision several practical applications of this work, including not limited to the following.
- It is widely reported that the first genome-wide Perturb-seq experiment cost $1M+ [2]. While costs are decreasing, new experiments must still be conducted for *each* new cell line (e.g. different cancers). Our model (CDN) offers a way to prioritize perturbations to screen for unseen cell lines, while learning from past screens.
- Despite clinical approval, a large number of drugs still lack [clear mechanisms of action](https://en.wikipedia.org/wiki/Category:Drugs_with_unknown_mechanisms_of_action). CDN is complementary to existing approaches for elucidating mechanisms and potential targets.

[2] Replogle et al. Mapping information-rich genotype-phenotype landscapes with genome-scale Perturb-seq. Cell. 2022.

---

### Meta-Review · Area_Chair_HFVV · 2024-12-20

**Metareview:**

The paper introduces a causal differential network for identifying intervention targets in single-cell perturbation data, combining causal discovery and an axial attention-based architecture. While addressing a relevant biological problem, reviewers have concerns about the limited methodological novelty, reliance on existing methods, and robustness to off-target effects and out-of-distribution data. Although the authors clarified design choices and provided detailed rebuttals, key issues regarding scalability, reliance on accurate causal graphs, and comparisons to simpler baseline methods remain unresolved. Overall, the work demonstrates potential but lacks sufficient innovation and clarity to justify its impact.

**Additional Comments On Reviewer Discussion:**

All reviewers participated in the discussion and acknowledged the rebuttal, but they remain concerned about the proposed method.

---

### Decision · Program_Chairs · 2025-01-22

Reject